



# Non Methane Hydrocarbon (C2-C8) sources and sinks around the Arabian Peninsula

Efstratios Bourtsoukidis[1], Lisa Ernle[1], John N. Crowley[1], Jos Lelieveld[1], Jean-Daniel Paris[2], Andrea Pozzer[1], David Walter[3], and Jonathan Williams[1]

[1]Department of Atmospheric Chemistry, Max Planck Institute for Chemistry, Mainz, 55128, Germany
[2]Laboratoire des Sciences du Climat et de l'Environnement, CEA-CNRS-UVSQ, UMR8212, IPSL, Gif-sur-Yvette, France
[3]Department of Multiphase Chemistry, Max Planck Institute for Chemistry, Mainz, 55128, Germany

*Correspondence to*: Efstratios Bourtsoukidis (e.bourtsoukidis@mpic.de)

**Abstract.** Atmospheric Non Methane Hydrocarbons (NMHC) have been extensively studied around the globe due to their importance to atmospheric chemistry and their utility in emission source and chemical sink identification. This study reports on shipborne NMHC measurements made around the Arabian Peninsula during the AQABA (Air Quality and climate change in the Arabian BAsin) ship campaign. The ship traversed the Mediterranean Sea, the Suez Canal, the Red Sea, the Northern Indian Ocean and the Arabian Gulf, before returning by the same route. This region is one of the largest producers of oil and gas (O&G); yet it is among the least studied. Atmospheric mixing ratios of C2-C8 hydrocarbons ranged from a few ppt in unpolluted regions (Arabian Sea) to several ppb over the Suez Canal and Arabian Gulf where a maximum of 166.5 ppb of alkanes was detected. The ratio between *i*-pentane and *n*-pentane was found to be $0.93 \pm 0.03$ ppb ppb$^{-1}$ over the Arabian Gulf which is indicative of widespread O&G activities, while it was $1.71 \pm 0.06$ ppb ppb$^{-1}$ in the Suez Canal which is a characteristic signature for ship emissions. We provide evidence that international shipping contributes to ambient C3-C8 hydrocarbon concentrations but not to ethane which was not detected in marine traffic exhausts. NMHC relationships with propane differentiated between alkane-rich associated gas and methane-rich non-associated gas through a characteristic enrichment of ethane over propane atmospheric mixing ratios. Utilizing the variability-lifetime relationship, we show that atmospheric chemistry governs the variability of the alkanes only weakly in the source dominated areas of the Arabian Gulf ($b_{AG} = 0.16$) and along the northern part of Red Sea ($b_{RSN} = 0.22$), but stronger dependencies are found in unpolluted regions such as the Gulf of Aden ($b_{GA} = 0.58$) and Mediterranean Sea ($b_{MS} = 0.48$). NMHC oxidative pair analysis indicated that OH chemistry dominates the oxidation of hydrocarbons in the region but along the Red Sea and the Arabian Gulf the NMHC ratios occasionally provided evidence for chlorine radical chemistry. These results demonstrate the utility of NMHCs as source/sink identification tracers and provide an overview of NMHCs around the Arabian Peninsula.



# 1 Introduction

Anthropogenic activities are estimated to be responsible for the release of 169 Mt of Non Methane Hydrocarbons (NMHCs) into the atmosphere each year (Huang et al., 2017). NMHCs are important in atmospheric chemistry as precursors of tropospheric ozone and particle formation, both of which have negative impacts on air quality, human health and climate

(Lelieveld et al., 2015; Seinfeld and Pandis, 2016; EEA, 2018). Ozone production is known to be efficient in Oil and Gas (O&G) basins (Edwards et al., 2014; Wei et al., 2014; Field et al., 2015) when the high NMHC concentrations react with the hydroxyl radical (OH) under low nitrogen oxide (NOx) regimes leading to carbonyls which photolyze to recycle the OH oxidant (Rohrer et al., 2014). Since ozone production is photochemical and catalysed by NOx, summertime urban pollution is frequently associated with elevated ozone concentrations (Parrish et al., 2004; Lelieveld et al., 2009; Perring et al., 2013;

Helmig et al., 2014).

Globally, anthropogenic NMHC sources are dominated by urban centres. Road transportation (including both evaporative and combustion emissions), residential combustion, transformation industry, fuel production and transmission, and solvent use contribute about three quarters to the estimated global total (Huang et al., 2017). Biomass burning (Andreae and Crutzen, 1997; Pozzer et al., 2010) in addition to biogenic sources (Poisson et al., 2000; Pozzer et al., 2010) and the Earth's degassing

(Etiope and Ciccioli, 2009) complete the diverse and dynamic inventory of NMHC sources. The dynamic nature of the NMHC source strength and hydrocarbon composition is driven by political, socioeconomic and environmental decisions. For example, reducing petrol and diesel powered vehicles can result in an overall decrease of transportation sector emissions but an increase in the share of alkanes in the summed emission NMHC (Huang et al., 2017). The general decline in fossil fuel usage had led to decreasing trends of global ethane concentrations (Aydin et al., 2011), but the demand for cleaner-than-coal burning energy

resulted in an expansion of natural gas production in the US (Swarthout et al., 2013) and reversal of global atmospheric ethane and propane trends due to US O&G activities (Helmig et al., 2016).

Since most sources have distinct emission composition patterns, hydrocarbon ratios are frequently used for identification and characterization of sources. The isomeric pentane ratio (*i*-pentane / *n*-pentane) has been used to characterise vehicle emissions, gasoline, O&G activities and urban conditions in general (Gilman et al., 2013; Thompson et al., 2014; Baker et al., 2008).

Correlations with propane, which is usually associated with natural gas processing and petroleum refining, can serve as an indicator for local/regional emission sources (Swarthout et al., 2013; Helmig et al., 2016) while the relative methane abundance can provide an indication of the type of gas (associated or non-associated with liquids) responsible for the emissions (Farry, 1998; Salameh et al., 2016).

Due to their relatively short atmospheric lifetimes that range from a few hours to several weeks, the variability of the NMHC

mixing ratios can serve to determine to what degree sources and sinks impact the measurements on a regional scale. Based on the Junge relationship (Junge 1974), Jobson et al. (1998) established a correlation between the natural logarithm of the standard deviation of the gas concentrations and the chemical lifetimes for shorter lived gases such as NMHC. This variability-lifetime relationship has been applied in characterising the impact of chemistry at various locations (Jobson et al., 1999;



Williams et al., 2000; Williams et al., 2001; Bartenbach et al., 2007; Helmig et al., 2008). In addition, the presence of specific oxidants can be detected by the use of NMHC oxidative pairs (Parrish et al., 2007; Baker et al., 2016; Young et al., 2014). This is particularly important for Cl radicals as there is no instrumentation available for direct Cl measurements, although measurements of $ClNO_2$ indicate a Cl source to be present.

The objective of this study is to present a comprehensive assessment of the atmospheric mixing ratios of the NMHCs that were measured in the marine boundary layer across the Mediterranean Sea and around the Arabian Peninsula during the AQABA ship campaign. NMHCs are used to characterise the various emission sources encountered (including ship, O&G, regional and urban emissions) and to evaluate atmospheric oxidation processes (by using lifetime-variability and selected oxidative pair relationships). Specific sources and sinks are studied by analysing NMHC mixing ratios in the extremely clean and polluted

conditions encountered within this rarely investigated region.

## 2 Methods

### 2.1 The AQABA ship campaign

To study the Air Quality and Climate in the Arabian Basin (AQABA) a suite of gas-phase and aerosol instruments were installed inside five air-conditioned laboratory containers and set on board the Kommandor Iona (KI) research vessel (IMO:

8401999, Flag: UK, Length Overall x Breadth Extreme: 72.55 m × 14.9 m). The shipborne measurement campaign lasted from the beginning of July to the end of August 2017, starting from the South of France, across the Mediterranean Sea, through the Suez Canal to Kuwait and back, covering around 20,000 km at sea (**Fig. 1**) at an average speed of $3.4 \pm 1.8$ m s$^{-1}$. When possible, in order to reduce sampling of its own exhaust, the ship sailed upwind of the shipping lane and at an angle to the prevailing wind. During the campaign, the average temperature was $30.6 \pm 3.9$ °C (range: $22.9 - 38.7$ °C) and the average

relative humidity was $71.4 \pm 16.3$ % (range $23.5 - 94.5$ %) while the wind speeds encountered were on average $5.4 \pm 2.8$ m s$^{-1}$ ( range: $0.7 - 14$ m s$^{-1}$).

### 2.2 Gas Chromatography – Flame Ionization Detector

#### 2.2.1 Instrumentation

Non methane hydrocarbons were measured with two coupled GC-FID systems (GC5000VOC and GC5000BTX; AMA

Instruments GmbH, Germany). The GC5000VOC was used for the quantification of light hydrocarbons ($C_2$-$C_6$) while the CG5000BTX was used for the heavier hydrocarbons and aromatics ($C_6$-$C_8$). Both systems share the same operating principle with sample collection on absorbent filled enrichment traps (Carbosieve/Carbograph 1:1 for AMA5000VOC and Carbotrap for AMA5000BTX), subsequent thermal desorption, chromatographic separation (Analytical column for GC 5000 VOC: AMA-sep Alumina, 0.32 mm ID, 50 m, 8 μm and for GC5000BTX: AMA-sep1, 0.32 mm ID, 30 m, 1.5 μm) and detection of the

eluting trace gases by flame ionization detectors. Their main difference is that the GC5000VOC system uses a dual-stage pre-



concentration principle, additionally equipped with a focussing trap (Carbosieve/Carbograph 1:1) and a stripper column (AMA-sep WAX, 0.32 mm ID, 30 m, 0.25 μm) in order to improve injection, peak resolution and chromatographic separation. To enable stand-alone combustion and carrier gas supply without the use of additional high pressure gas cylinders, a hydrogen generator (H2PD-300-220, Parker Hannifin Corporation, USA) and zero air generator (AK6.1, Innotec GmbH & Co.KG, Germany) were operated in parallel. The zero air generator was supplied with compressed air (Compressor UA-025K/05641100, Dürr Technik GmbH & Co.KG, Germany) at 2 bar, after three-stage filtering through cartridges filled with different adsorbent materials. The first cartridge is filled with silica gel for drying, the second with charcoal for ozone and particle removal, and the third with a mixture of charcoal, molecular sieve and soda lime for $CO_2$ and organic trace gas removal. Inside the zero air generator, a palladium catalyst is heated to 430 °C to eliminate any remaining hydrocarbons and carbon monoxide. The zero air produced was used for both FID flames in addition to the Nafion drier.

### 2.2.2 Water and ozone interferences

The necessity of water and ozone removal in NMHC quantification has been frequently reported in the literature (Helmig and Greenberg, 1995; Plass-Dülmer et al., 2002; Slemr et al., 2002; Apel et al., 2003; Rappenglück et al., 2006). As the campaign was planned for the moist marine boundary layer and for areas that were likely to have high ambient ozone concentrations (Lelieveld et al., 2002; Lelieveld et al., 2009), the effects of humidity and ozone were studied in the laboratory prior to deployment.

Moist air (relative humidity = 100 %) substantially influenced ethene concentrations when compared to dry air (120 % difference) while it had a lesser effect on propene (16% difference), benzene (28 % difference), and all other species (<10 %). The use of the Nafion drier (Perma Pure LLC) effectively eliminated these discrepancies as it proved capable of drying the ingoing sample air without producing chromatographic artefacts for the NMHCs investigated.

Stepwise addition of $O_3$ up to mixing ratios of 200 ppb showed no significant impact of ozone on the alkane measurements. The most affected species were the alkenes and in particular isoprene which decreased in mixing ratio by 25 % per 100 ppb of $O_3$. Therefore two ozone scrubbers were tested. The potassium iodine scrubbers (LpDNPH, Supelco Analytical, USA) showed limited $O_3$ removal capacity (ca. 45 L at 70 ppb $O_3$) and induced chromatographic artefacts such as noisy background and ghost peaks. In contrast, $Na_2S_2O_3$ impregnated quartz filters removed $O_3$ with much larger capacity (>200 L at 70 ppb $O_3$) without affecting the quality of the chromatograms. We therefore used the $Na_2S_2O$ infused quartz filters for ozone removal in the NMHC sampling.

### 2.2.3 Sampling

The GC-FID system was installed inside one of the five air-conditioned laboratory containers. A 5.5 m tall (3 m above the container), 0.2 m diameter, high flow ($\approx$ 10 m$^3$ min$^{-1}$) stack was used as the common inlet for all gas phase measurements. For NMHC sampling, a sub-flow of 2.5 L (stp) min$^{-1}$ (Lpm) was drawn through a PTFE filter (5 μm pore size, Sartorius Corporate Administration GmbH, Germany) for particle removal and then through a heated (ca. 40 °C) and insulated PFA-Teflon line



(12 m length, OD = 0.635 cm) with a residence time of approximately 9 sec. The PTFE filter was exchanged every 2-5 days depending on the aerosol and sea salt load encountered. Inside the container, the GC-FID systems sampled air from the main 2.5 Lpm stream at a rate of 90 sccm (2 x 45 cm$^3$ (stp) min$^{-1}$ (sccm), through an ozone scrubber ($Na_2S_2O$ infused quartz filters) and a Nafion dryer (500 sccm counter-flow) that were used to eliminate the effects of ozone and humidity in sample collection. Similar to ambient samples, the calibration gas was passed through the ozone scrubber and Nafion dryer to ensure identical calibration conditions through the complete campaign (**Fig. S1**).

The broad range of NMHC concentrations encountered along the ship track dictated in-situ adjustments in the sampling volume. It is worth noting that in addition to ambient concentrations, high waves that cause intense yaw, pitch and roll movements of the ship can disturb flow controllers, influence the flame stability and signal baseline and hence detection limits. Therefore, depending on ambient NMHC concentrations and wave conditions, three sampling volumes were used throughout the campaign. In polluted conditions, such as the Arabian Gulf and the Suez Canal, short sampling times (10 min) and volumes (450 ml) allowed higher time resolution (50 min per measurement), while under clean conditions, such as found in the Arabian Sea, longer sampling times and volumes (30 min, 1350 ml, time resolution = 1 h) improved detection limits. For most of the route, the sampling time was 20 min, the sampling volume 900 ml, and time resolution 50 min per measurement.

Temperature control of the traps is done by Peltier elements which permit rapid heating and cooling rates along with high temperature reproducibility. Sample collection was carried out at 30 $^o$C for all traps, while desorption temperature was at 230 $^o$C. The typical container temperature was about 25 $^o$C and condensation was avoided with the use of heated lines and the Nafion drier. A different oven temperature program of 38 min cycle was selected for the two GC-FID systems. The initial oven temperature was set to 50 $^o$C and 60 $^o$C while the final oven temperature was 160 $^o$C and 200 $^o$C for the GC5000VOC and GC5000BTX systems respectively. The detailed temperature program for each GC-FID system can be seen in **Fig. S2**.

### 2.2.4 Calibrations

Frequent calibration and blank samples ensured optimum and stable performance of the GC-FID systems. A multicomponent reference gas mixture (National Physical Laboratory, UK, 2015) was diluted with synthetic air (6.0, Westfalen AG, Germany) and passed through both the ozone scrubber and the Nafion drier to provide sampling conditions similar to ambient air. In total, 79 calibration and 20 blank samples were obtained throughout the campaign. Prior to ambient samples, a 6 step calibration curve (6 x 3) was obtained for 450 ml and 900 ml sampling volume, confirming the high sensitivity and excellent linearity of FID detectors as the $R^2$ of the linear fit for all species was always higher than 0.98. During the campaign, single calibration points and blanks were frequently obtained to monitor the sensitivity while minimising data loss. Analysis of the initial calibrations revealed volume dependent sensitivity for ethane, propane, ethene and propene. Therefore, another set of linear calibrations with 1350 ml sampling volume was performed towards the end of the campaign. For the aforementioned species, the volume dependent calibration curve was applied for deriving the ambient mixing ratios.




Due to varying baseline noise, dependent on the ship motion, condition specific detection limits (DL) and total uncertainties have been calculated (**Table 1**). The limit of detection was calculated from the baseline noise, as it was determined by IAU Chrom intergration software (Sala et al., 2014). Peaks that were higher than 3 times the noise signal were considered to be above DL. With this approach DLs for each calibrated chromatograph were derived. Similarly, a total uncertainty for each

sample was derived. For the total uncertainty we propagated 5 % uncertainty in the measured flows combined with the % inherent reference gas mixture uncertainty and the varying peak uncertainty that was calculated as the percentage signal to noise ratio for each chromatographic peak.

### 2.2.5 Ship exhaust filter

During the complete ship campaign 1393 samples were collected by the GC5000VOC system and 1244 samples by the

GC5000BTX system, resulting in 26,668 integrated peaks. For the purpose of this study, all samples that have been obtained in ports or under stationary conditions have been excluded from the analysis with the only exception being the pentane mixing ratios measured in Jeddah port (JP). Similarly, samples influenced by the KI ship exhaust were considered only for the NMHC composition characterization and excluded from the ambient air assessments. To identify which samples were affected by our own ship exhaust, a combination of wind direction relative to the ship movement and measured $NO_2$, $O_3$ and $SO_2$ were used to

create a flag for affected timeframes. Since ethene mixing ratios were exceptionally high inside the exhaust plume, as has been reported previously (Eyring et al., 2005), all flagged samples were individually inspected for ethene, resulting in a filter that excluded all samples that were contaminated by the emissions of KI ship exhaust. Ethane was clearly not emitted by KI ship exhaust and therefore the filter was not applied for this particular hydrocarbon. The number of samples quantified per measured species can be seen in **Table 2**.

**2.3 Methane measurements**

The CH4 molar mixing ratio was measured using a Cavity ringdown spectroscopy analyser (Picarro G2401). Four different calibration gases traceable to WMO X2004A scale and bracketing typical ambient concentrations were injected in the analyser at ports and every 15 days for calibration purposes. The injection sequence consists of four 15-min injections of each of the four gases. An additional target gas was injected daily to assess measurement accuracy. The data has been documented and

processed following ICOS (Integrated Carbon Observing System) standard procedure (Hazan et al., 2016), including the propagation of the calibration and threshold-based filters. The mean drift of measured concentrations from calibration cylinders between two sequences is 0.5 nmol mol$^{-1}$, significantly below the drifts typically observed at fixed observatories (Hazan et al., 2016).

**2.4 ECHAM5/MESSy Atmospheric Chemistry (EMAC) model**

The ECHAM/MESSy Atmospheric Chemistry (EMAC) model is a numerical chemistry and climate simulation system that includes sub-models describing tropospheric and middle atmosphere processes and their interaction with oceans, land and





human influences (Jöckel et al., 2010). It uses the second version of the Modular Earth Submodel System (MESSy2) to link multi-institutional computer codes. The core atmospheric model is the 5th generation European Centre Hamburg general circulation model (ECHAM5, Roeckner et al., 2006). For the present study we applied EMAC (ECHAM5 version 5.3.02, MESSy version 2.53.0) in the T106L31-resolution, i.e. with a spherical truncation of T106 (corresponding to a quadratic

Gaussian grid of approx. 1.1 by 1.1 degrees in latitude and longitude) with 31 vertical hybrid pressure levels. A complex organic chemistry (MOM, Mainz Organic Mechanism) is integrated in the model as described in Sander et al. (2018). The simulation set-up is analogous to the one of Lelieveld et al. (2017).

## 2.5 Hybrid Single Particle Lagrangian Integrated Trajectory (HYSPLIT) model

Back-trajectories of air parcels encountered have been calculated with the Hybrid Single-Particle Lagrangian Integrated

Trajectory model (HYSPLIT, version 4, 2014), which is a hybrid between a Lagrangian and an Eulerian model for tracing a small imaginary air parcel forward or back in time. The model can be accessed under https://ready.arl.noaa.gov/HYSPLIT.php while further details are provided in (Draxler and Hess, 1998). For the purposes of this study, back-trajectories with a start height of 200 m above sea level have been calculated, starting at the ship position and going back in time on an hourly time grid (**Fig 1**).

**3. Results and discussion**

### 3.1 Overview of atmospheric mixing ratios

Widely varying mixing ratios of alkanes, alkenes and aromatics were quantified along both legs of the ship campaign (**Fig. 2, Figs. S3 − S19**). To investigate regional variations in atmospheric chemistry and abundance of NMHCs, the track was subdivided in eight regions: Mediterranean Sea (MS), Suez Canal (SC; including Great Bitter Lake and Suez Gulf), Red Sea

North (RSN), Red Sea South (RSS), Gulf of Aden (GA), Arabian Sea (AS), Gulf of Oman (GO) and Arabian Gulf (AG) (for coordinates see **Table 2**). Despite the leg-to-leg differences in the origin of the air masses (**Fig. 1**), ethane mixing ratios match well with this geographical demarcation.

The cleanest air masses were measured over the Arabian Sea (AS) where the majority of NMHCs were below or close to the detection limit (**Table 2**). HYSPLIT back-trajectories (**Fig. 1**) show that these air masses sampled originated from the Indian

Ocean and were transported to the sampling point in the prevailing largescale anticyclonic system. Land influence was confined to a small part of the eastern Somalian desert, where NMHC sources are negligible. Mean ethane mixing ratios over the AS were $260 \pm 99$ ppt on average, with the lowest recorded value being 155 ppt; an indicative value for the regional tropospheric background. Despite the small variability in comparison to the majority of the NMHCs measured (**Fig. 3**), ethene was found to be relatively high on average ($90 \pm 57$ ppt) with a maximum recorded value of 240 ppt. As ethene is highly

reactive towards atmospheric oxidants and therefore has short atmospheric lifetime (several hours), high mixing ratios are indicative of a local source. Ship traffic is unlikely to be the primary source of ethene in the AS region because of the low



abundance of other ship emission related NMHCs. Therefore, we hypothesise that the increased ethene mixing ratios are emissions from marine phytoplankton as has been previously reported from other locations (Plass-Dülmer et al., 1995; Plettner et al., 2005).

Similar conditions were met over the Mediterranean Sea (MS) where ethene was 114 ± 43 ppt on average, and with a 204 ppt
maximum. Nonetheless, average NMHC mixing ratios over the MS were generally higher compared to the AS, presumably as upwind European sources were closer to the sampling point over the MS than Southeast Asian sources over the AS. Interestingly, the variability-lifetime dependency (Jobson et al., 1998; Williams et al., 2000) was more pronounced for MS (see section 3.3) suggesting more biogenic input in the AS. The rest of the alkenes and higher alkanes were frequently below the detection limit in the MS region and the average values presented in **Table 2** are mainly driven by the elevated mixing
ratios measured close to southern Italy.

Mixing ratios were markedly different in the Suez Canal and the Gulf area (SC). The increased marine traffic intensity in combination with the proximity to populated areas along the Suez Canal resulted in wide-ranging NMHC mixing ratios (**Fig. 3**). The most abundant hydrocarbon was propane (3.9 ± 6.15 ppb), followed by *n*-butane (3 ± 0.56 ppb), ethane (2.64 ± 2.93) and *i*-butane (1.39 ± 2.24 ppb) (average values). Accounting for both isomers, butane was the dominant NMHC for about 50
% of the samples that were collected inside SC (**Fig. S20**). Alkenes were dominated by ethene (0.81 ± 1.11 ppb) with the maximum recorded value of 5.59 ppb, but propene, trans-2-butene and 1-butene were also present in substantial amounts. The area of Suez had in general the highest aromatic hydrocarbons measured throughout the campaign since benzene (160 ± 183 ppt), toluene (276 ± 447 ppt) and *m*-, *p*- xylenes (107 ± 164 ppt) were measured as high as 0.73, 1.69 and 0.61 ppb, respectively.

Interestingly, the northern part of Red Sea (RSN) displayed contrasting NMHC signatures compared to its southern counterpart (RSS). The air that was encountered in RSN originated from the northeast part of Africa (Egypt, Libya), with vestigial influences from the south-eastern European continent. Alkane mixing ratios were dominated by ethane and propane, followed by butanes and pentanes, generally displaying a decrease of mixing ratio with higher carbon numbers. Depending on the wind direction, local sources delivered high mixing ratios of ethane (max = 17.33 ppb) and propane (max = 10.45 ppb), but
in relation to the SC, lower butanes and pentanes. For instance, the SC area had an average propane to butane ratio of 0.88 while the respective ratio in RSN was 1.5 due to the different types of sources (see section 3.2).

The air sampled at the southern part of the Red Sea (RSS) was mainly influenced by central Africa, with 4 day back-trajectories extending towards the desert areas of Sudan and Chad. Nonetheless, a substantial number of samples originated from along the east coasts of Sudan and Egypt with influences from the RSN region. Data distribution plots (**Fig. S20**) show
that the benzene to toluene ratio was higher than unity for about 60 % of the samples and below one for the rest, indicating influences by fresh ship emissions and possibly biomass burning. Ethane mixing ratios varied relatively little (668 ± 297 ppt) due to the lack of O&G activities in the RS area, in addition to the negligible input from marine traffic. In contrast, marine traffic associated gases were detected in significant quantities with the majority of NMHCs ranging from few ppt to orders of magnitude higher mixing ratios.



The GA region showed similar atmospheric mixing ratios to both RSS and AS, and the back-trajectories suggest common influences with the aforementioned areas. The majority of the samples had mixing ratios in the sub-ppb range with the isomeric sum of butanes (75 ± 218 ppt) and pentanes (89 ± 179 ppt) being very similar (**Fig. S20**) and always lower than ethane (438 ±121 ppt) and propane (166 ± 174 ppt).

In contrast, the GO region was dominated by C2-C5 alkanes that were all measured to be in approximately the same concentration range. The most abundant NMHC measured was ethene (13.33 ppb) that in combination with the increased ethane (546 ± 391 ppt), propane (551 ± 564 ppt), butanes (570 ± 638 ppt) and pentanes (438 ± 482 ppt) indicated direct influences from O&G production in addition to urban pollution and marine traffic. As the air masses encountered in the GO displayed similar back-trajectories to those from AS, the increased concentrations can be mainly attributed to anthropogenic

emissions from Oman that include O&G production.

Over the AG, an area of intense O&G activity, C2-C5 alkanes were on average higher by an order of magnitude ($AG_{ethane}$ = 7.82 ± 9.98 ppb, $AG_{propane}$ = 7.93 ± 10.5 ppb, $AG_{butanes}$ = 7.09 ± 9.81 ppb, $AG_{pentanes}$ = 2.97 ± 4.08 ppb) again displaying wide-ranging mixing ratios (**Fig 2, Fig. 3, Fig. S20**). Similarly, alkenes and aromatic hydrocarbons were highly abundant ($AG_{ethene}$ = 0.63 ± 0.86 ppb, $AG_{propene}$ = 0.08 ± 0.11 ppb, $AG_{benzene}$ = 0.12 ± 0.07 ppb, $AG_{toluene}$ = 0.05 ± 0.05 ppb, $AG_{xylenes}$ = 0.09 ± 0.08

ppb) but on average lower than the mixing ratios encountered in SC. Due to instrumental problems the GC5000BTX was not operating at the northern part of AG, the area with the highest observed C2-C5 mixing ratios. The high variability observed in this area is attributed to the diverse influence of local strong O&G sources. Fresh emissions from oil fields were sampled over the northern part of AG and contained large amounts of alkanes ($AG_{ethane,max}$ = 48.02 ppb, $AG_{propane,max}$ = 53.79 ppb, $AG_{butanes,max}$ = 41.26 ppb, $AG_{pentanes,max}$ = 23.45 ppb), but occasionally the sampled NMHCs (in particular during the second leg

of the campaign) were already oxidized, in relatively clean and dry air originating from the Iranian dessert (**Fig. 1b**). Ethane and propane were measured as low as 0.4 and 0.31 ppb, respectively, at the time while alkenes were in the sub-ppt level. Despite the variability in air mass origin between the two legs, the samples provide a distinct signature of the area which is further investigated in detail in section 3.2.

Overall, the background latitudinal gradient of ethane and propane mixing ratios (Blake and Rowland, 1986; Simpson et al.,

2012) was only apparent when considering the minimum mixing ratios in each region (**Fig. 4**). The Middle East is a global hot-spot of ethane and propane emissions as fossil fuel emissions are highly concentrated due to regional petrochemical activities. Hence, the mixing ratios that were measured along SC, RSN and AG are characteristic of the local emission sources and they cannot be used to derive a latitudinal gradient. Nonetheless, ethane mixing ratios along the AS can be considered as unaffected by anthropogenic activities, displaying a decline of 4.6 ppt per latitudinal degree. Concerning propane, the situation

is different as marine traffic emissions influence regional atmospheric mixing ratios and hence higher variability in their abundance was observed in AS and in most of the investigated areas.



### 3.2 Source identification through NMHCs

#### 3.2.1 Pentane isomers

Typical anthropogenic sources (e.g. O&G operations, gasoline vapours, vehicle emissions, urban areas) have distinct NMHC signatures and their mixing ratios can be used for identification and characterization of the respective emission source. The relationship between pentane isomers had been frequently used for emission-source identification as natural gas (Gilman et al., 2013; Swarthout et al., 2013; Thompson et al., 2014), gasoline vapours (Gentner et al., 2009), vehicle emissions (Harley et al., 1992; Broderick and Marnane, 2002) and in general urban environments (Warneke et al., 2007; Baker et al., 2008; Von Schneidemesser et al., 2010; Barletta et al., 2017; Panopoulou et al., 2018) display a distinct relationship between *i*-pentane and *n*-pentane. Higher *i*-pentane is associated with fuel evaporation (McGaughey et al., 2004) and vehicle emissions (Jobson et al., 2004) while *n*-pentane is more abundant in natural gas (Gilman et al., 2013). Since the *i*&*n*- pentane isomers have approximately the same reaction rate with the hydroxyl radical (Atkinson, 1986), their atmospheric oxidation proceeds at equivalent rates and their ratio is a reliable marker for identifying emission sources independent of their proximity to the sampling point.

The relationship between pentane isomers is commonly expressed with the use of enhancement ratio (ER) which is defined as the slope term in a linear regression between *n*-pentane and *i*-pentane. **Fig. 5** compares the values between pentane isomers for three sampled regions (JP, SC, AG) together with enhancement ratio slopes (ER) that are reported in the literature. The samples that were collected in Jeddah port show the highest ratio of *i*- to *n*- pentane ($ER_{JP}$ = 2.9 ppb ppb$^{-1}$, $R^2$ = 0.96), with the slope being on the upper range of urban conditions and almost identical with the slope that is representative of vehicle emissions. During the two day stop in Jeddah port, the ship was anchored in a part of the port with considerable activity from land vehicles inside the port, possibly explaining the observed ER. In addition, the prevailing north / northeast winds advected air from the nearby city of Jeddah, an urban centre with population of ca. 4 million habitants. Barletta et al. (2017) reported a ratio of 2.66 ppb ppb$^{-1}$ for the city of Jeddah which agrees with our observations.

Along Suez, the pentane enhancement ratio slope was $ER_{SC}$ = 1.71 ppb ppb$^{-1}$ ($R^2$ = 0.98). SC air was predominantly influenced by ship traffic and potentially by urban emissions from the cities that are distributed along the channel. In addition, O&G operations along the Great Bitter Lake and Gulf of Suez, with occasional oil rigs along the route, comprise a mixture of emission sources that could affect the isomeric pentane abundance. However, the slope is identical to the one that was observed in the Texas ship channel (1.7 ppb ppb$^{-1}$; Blake et al., 2014). By using all filtered samples that were directly impacted by KI exhaust, we derive an enhancement ratio of 1.59 ppb ppb$^{-1}$ ($R^2$ = 0.99). While this ratio is affected by regional ambient concentrations and the presence of KI exhaust sampled volume, its values, the high linearity of fit and the ratio slopes obtained in SC and reported for the Texas ship channel indicate this is a characteristic signature of ship emissions. Interestingly, C2-C5 alkanes are not considered to be emitted by international shipping (Eyring et al., 2005). While we did not observe any ethane emissions from the KI exhaust, propane, butanes and pentanes constituted about 50% of the composition



of a typical KI exhaust sample (**Fig. 6)**. Given the obvious similarities in pentane isomer relationships when comparing KI exhaust and SC samples, it is evident that ship emissions strongly impact the hydrocarbon mixture in the SC.

The intense O&G operations in the AG area could be discerned by another characteristic relationship between pentane isomers. The enhancement ratio ($ER_{AG}$ = 0.93 ppb ppb$^{-1}$, $R^2$ = 0.97) is very similar to the natural gas signature (ER = 0.86 ppb ppb$^{-1}$; Gilman et al., 2013) with almost all data points falling within the O&G range (± 20%). Comparable values have been reported for the oil and gas rich northeastern Colorado, including Boulder (ER = 1.1 ppb ppb$^{-1}$; Gilman et al., 2013), Fort Collins (0.809 ppb ppb$^{-1}$; Gilman et al., 2013), Erie/Longmont wells (ER = 0.965 ppb ppb$^{-1}$; Thomson et al., 2014) and Boulder Atmospheric Observatory (ER = 0.885 ppb ppb$^{-1}$; Gilman et al., 2013, ER = 1 ppb ppb$^{-1}$; Swarthout et al., 2013), as well as for the natural gas activities on Kola Peninsula in Russia (ER = 1.1 ppb ppb$^{-1}$; (Gilman et al., 2010)).

Despite the high correlation between pentane isomers in the AG, several samples and in particular five samples with *i*-pentane > 1 ppb, deviate substantially from the derived slope (**Fig. 5**), displaying a higher *i*- / *n*- pentane ratio. Six day HYSPLIT back-trajectories show that these particular samples were influenced by the large-scale oil fields (US-CIA, 2007) and gas flaring areas (https://skytruth.org/viirs/) in addition to major cities in Iraq (**Fig. 7**). Before and after encountering the plume of high *i*- to *n*- pentane ratios, the back-trajectories extend to the gas fields of Turkmenistan before crossing Iran and reaching the sampling point. Apart from the distinctly high *i*- / *n*- pentane ratio, the common feature of all samples that originated from the Iraqi oil fields is the high *n*-butane and propane mixing ratios that reached values up to 26.88 and 53.79 ppb respectively.

### 3.2.2 Correlation with propane mixing ratios

Elevated propane mixing ratios are usually associated with natural gas processing and petroleum refining. Light alkanes, such as ethane, *n*-butane and *n*-pentane, are co-emitted by O&G activities and due to their short lifetimes, a tight correlation with propane indicates common sources and little photochemical processing. Therefore, propane is frequently regarded as a sensitive indicator for local/regional emission sources (Helmig et al., 2016).

Weak propane-NMHC correlations were generally observed over the MS, RSS, GA, AS and GO despite occasional co-variation related to specific sources. In these areas, propane mixing ratios were relatively low with median values ranging from 107 ppt over the AS to 355 ppt at GO (**Table 2**). The higher mixing ratios that were measured over the GO are associated with influences by O&N activities in Oman, but the number of samples with elevated propane mixing ratios was outnumbered by the samples originating from the Arabian Sea. Considering the complete propane dataset, the aforementioned regions displayed only moderate correlations with the longer lived ethane (0.47 < $R^2$ < 0.72) and poor correlations with the rest of the alkanes, alkenes and aromatics.

The three regions that showed a high correlation between propane and NMHCs were SC, RSN and AG (**Fig. 8, Table 3**). Among them, the propane-NMHCs relationship was markedly different in the SC as propane mixing ratios were correlating only with butanes, alkenes and aromatic hydrocarbons. *N*-butane in particular was the only hydrocarbon that was emitted more strongly than propane (ER = 1.1 ppb ppb$^{-1}$, $R^2$ = 0.91) while the known tight relationship with its isomer *i*-butane resulted in an enhancement mixing ratio slope of 0.43 ppb ppb$^{-1}$ ($R^2$ = 0.89) for *i*-butane. The SC was the only region that had high





propane correlations with alkenes ($0.73 < R^2 < 0.81$) and aromatic hydrocarbons ($0.87 < R^2 < 0.91$), with ethene and toluene displaying the highest enhancement ratios towards propane with values of 0.22 and 0.24 ppb ppb$^{-1}$ respectively. The co-emission with combustion tracers, such as aromatic hydrocarbons and the highly reactive alkenes, provide further evidence that fresh marine traffic emissions dominate NMHC abundance in this area. Finally, the poor relationship between propane

and ethane in SC is in agreement with our ship exhaust observations, showing that ethane is not emitted by ships.

In contrast to SC, RSN and AG measurements displayed high correlation coefficients with ethane that was found at even higher mixing ratios than propane in RSN (ER = 1.3 ppb ppb$^{-1}$, $R^2 = 0.95$). A similar enhancement ratio (1.5 ppb ppb$^{-1}$) has been reported by Swarthout et al. (2013) from the Boulder Atmospheric Observatory (BAO) within a wind sector representative of natural gas sources. Likewise, butane and pentane isomers showed high correlations with propane and with

enhancement ratio slopes (**Table 3**) that are very similar to those observed in BAO (Pétron et al., 2012; Gilman et al., 2013; Thompson et al., 2014; Swarthout et al., 2013; Pétron et al., 2014). Benzene and toluene also correlated with propane but with moderate correlation coefficients as the fit is mainly affected by the larger variability of these species in the southern part of RSN. Overall, the relationships between propane and the rest of the hydrocarbons measured provide indications for a natural gas source in the region that is not documented in O&G cartographic maps (US-CIA, 2007). Etiope and Ciccioli (2009) have

shown that the RSN has a substantial coverage of geothermal and volcanic spots that could potentially explain the high mixing ratios of NMHCs measured in this region.

The Arabian Gulf is a global hot-spot for O&G activities. The correlations of alkanes with ambient propane mixing ratios were very similar with the ones from RSN, with the main difference being that ethane was emitted at lower rates (ER = 0.91 ppb ppb$^{-1}$, $R^2 = 0.9$) and *i*-butane at higher rates (ER = 0.52 ppb ppb$^{-1}$, $R^2 = 0.83$). Aromatic hydrocarbons did not correlate well

with propane in AG, where the aromatic hydrocarbon measurements are in the low range due to missing measurements along the northern, more polluted part of AG (see **Fig. S17-S19**). Despite the generally high correlation between propane and the rest of the alkanes in AG, the highest propane mixing ratios ([$C_3H_8$] > ~10 ppb) had slightly different interrelationships with all hydrocarbons, which is attributed to a different air-mass origin (and therefore source), as seen for example in the case study of **Fig. 7**.

**3.2.3 Excess mole fraction**

To further investigate the different interrelationships observed under high propane mixing ratios in the AG, and to generally provide a quantitative characterization of hydrocarbons in each region, their relative abundance was calculated. We define as "excess NMHC mole fraction" (EMF) the relative abundance of non-methane hydrocarbons to the sum of methane and non-methane mixing ratios, whereby the background mixing ratios (defined as the lowest 1% of regional samples) of each area

were subtracted from the total mixing ratios, resulting in the following equation:

$$Excess\ NMHC\ mole\ fraction\ (EMF) = \frac{\sum(NMHC - NMHC_{BG})}{([CH_4] - [CH_4]_{BG}) + \sum(NMHC - NMHC_{BG})} * 100\% \qquad (1)$$



While methane on the global scale is primarily emitted by wetlands, agriculture and waste sources, natural gas and petroleum systems contribute up to a quarter of global anthropogenic emission, and 10% of the global source (Warneck and Williams, 2012; Kirschke et al., 2013; Saunois et al., 2016). Non-associated natural gas (produced by reservoirs without liquid)

composition is dominated by methane while associated gas contains smaller fractions of methane and higher of alkanes (Farry, 1998; Matar and Hatch, 2001; Anosike et al., 2016). **Fig. 9** shows the propane versus ethane mixing ratios in AG, coloured by the excess NMHC mole fraction. Propane and ethane display different relationships under high EMFs, where ethane is higher than propane with an enhancement ratio slope of 1.1 ppb ppb$^{-1}$ when considering EMF > 20 %. Similar ratios have been obtained through the use of PMF (Positive Matrix Factorization) analysis for oil refineries in Houston (Kim et al., 2005;

Buzcu and Fraser, 2006) and with measurements over an oil well (Colombo et al., 2013).

On the 29$^{th}$ of July 2017 at 12:00 UTC, KI crossed an oil slick in the AG over which a 10 minute sample was collected. This particular sample contained 30.1 ppb of C2-C8 hydrocarbons, consisting mainly of alkanes and aromatics while the mixing ratios of alkenes remained at similar levels to the samples before and after the oil slick. The mole fraction of this sample (21 %) and its position on the respective linear fit (**Fig. 9**) indicate that associated gas contains a larger fraction of NMHCs.

Higher EMFs were assigned to back-trajectories originating from the oil fields and refineries of Iran while the lower EMFs were associated with trajectories originating from the gas fields of Turkmenistan (see notably the case study of **Fig. 7**). Our analysis supports the notion that associated gas contains more non-methane hydrocarbons than non-associated gas, and that EMF can therefore be used to distinguish between emissions related to oil versus gas exploitation. We find also that the relationship between ethane and propane can be used to identify the type of emission source. Nonetheless, this observation is

valid only for relatively fresh emissions as the shorter propane lifetime will modify the ethane to propane ratio in chemically processed air masses.

All regions occasionally yielded samples with relatively high (to the regional medians) EMFs (**Fig. 10**) that were possibly related to oil slicks or oil production activity sources. A prominent example is the GO. Besides AG (12 ± 13.7 %) the northern part of Red Sea had, with a median EMF value of 11 ± 6.6 %, the smallest regional standard deviation calculated. This

observation is indicative of a widespread and relatively homogenous source of NMHCs in the area as the sum of alkanes was on average the second highest encountered along the sampled regions. Pentane isomers (section 3.2.1) and relationships with propane (section 3.2.2) indicated that ship emissions dominated the NMHC abundance in SC. The high correlation coefficients between methane and the majority of NMHCs in combination with the poor correlation with ethane in this area (**Table 3**) confirm that the EMF in SC is primarily driven by marine traffic; a known source of both methane and non-methane

hydrocarbons (Eyring et al., 2005) but with varying relationships depending on the ship type and fuel used (Anderson et al., 2015). The areas of GA and AS were dominated by CH$_4$ (EMFs < 5 %) which was frequently close to global background levels. In the absence of regional emission sources, the more reactive NMHCs are removed by atmospheric radicals, and the longer-lived methane, which has a large oceanic source (e.g. Bange et al., 1998, Born et al., 2017), dominates the mole





fraction. Therefore, these areas are more appropriate for investigation of atmospheric oxidation processes better than the identification and characterization of emissions.

### 3.3 Sink characterization through NMHCs

### 3.3.1 Variability-lifetime relationship

The relationship between the variability of atmospheric mixing ratios and their chemical lifetimes can be used to evaluate the relative importance of chemical reactions and source proximity in the observed variability of trace gases (Junge, 1974; Jobson et al., 1998; Jobson et al., 1999; Williams et al., 2000; Williams et al., 2001; Pszenny et al., 2007; Helmig et al., 2008). The conceptual approach has also been extended to particles (Williams et al., 2002). Jobson et al. (1998) introduced the standard deviation of the natural logarithms of the mixing ratios as a useful measure for the variability for short lived species. The

relationship with chemical lifetime follows a power law distribution of the form:

$$S(\ln(X)) = A \cdot \tau^{-b} \tag{2}$$

where,

$$\tau = \frac{1}{k_{OH,X}[OH] + k_{Cl,X}[Cl] + k_{NO3,X}[NO_3]} \tag{3}$$

Here, S is the standard deviation of the natural logarithm of hydrocarbon X with a calculated chemical lifetime ($\tau$) with respect to reaction with the hydroxyl [OH], chlorine atoms [Cl] and the nitrate [$NO_3$] radicals. A and b are fitting parameters that are

determined by the regression. The term A has been interpreted (although interpretations vary) as related to the standard deviation of the air mass transport times while b is a robust indicator of the dependence of the measured variability on atmospheric chemistry (as opposed to source proximity) and hence remoteness from sources (Jobson et al., 1998). The relationship has also been exploited in the past to derive gas and particle lifetimes, to determine data quality, and to estimate radical abundances (Williams et al., 2000). In general, higher variability is expected for the most reactive species (Junge,

1974). The parameter b ranges from 0 (indicating close-by sources and no dependence on chemistry) to around 0.6 (a value that is typical of remote locations in the troposphere).

The chemical lifetime depends on the respective reaction rate coefficients and on the OH, $O_3$, $NO_3$, and Cl concentrations. For our calculations we use the average regional radical concentrations as they have been calculated by the EMAC-MOM chemistry model (**Table 4**), combined with the reaction rates shown in **Table 5**. We chose to investigate the relationship solely

with C2-C5 alkanes as these species were above the detection limit for all areas and therefore provide a robust measure for comparing different regions. They also do not react rapidly with ozone so this can be neglected. In addition, we found that



reactions with the nitrate radical ($NO_3$) are negligible (i.e. do not affect the calculated lifetimes; <1% changes) due to their low rate coefficients with C2-C5 alkanes (Atkinson, 1991; http://iupac.pole-ether.fr/ ).

**Fig. 11** shows that the smallest b value was derived for AG (0.16 ± 0.09); a region where adjacent O&G activities dominated the emissions resulting in increased and highly variable alkane mixing ratios (25.44 ± 33.33 ppb). Due to the source proximity,

C2-C5 hydrocarbon variability was not significantly related to the chemical reactions. Similarly, the small $b_{RSN}$ (0.22 ± 0.11) is also indicative of nearby sources so that wind direction rather than oxidation time determines atmospheric variability. RSN displayed smaller standard deviations and higher atmospheric lifetimes for all species. Both $b_{AG}$ and $b_{RSN}$ fall in the lowest range of b values that are consistent with nearby emission sources.

SC and GO displayed very similar b factors from the variability-lifetime relationships, and were both higher than found in AG.

The highest variability along the route was encountered in the SC which is also the area with the modelled highest OH and Cl concentrations, resulting in shorter lifetimes. Therefore, the $b_{SC}$ (0.33 ± 0.27) represents a combination of regional sources and oxidation processes. Similarly, the GO ($b_{GO}$ = 0.28 ± 0.23) had high OH but low Cl concentrations according to the model and mixing ratio variability. In this case the standard deviation of the most reactive pentanes drives the fit. Considering AS, the poor regression fit ($R^2$ = 0.33) does not allow interpretation of the derived parameters but it is indicative for local biogenic

sources as the variability-lifetime relationship is not as clear as expected for remote sources.

The areas with the highest dependence of variability on chemistry were found to be the GA, MS and RSS. In these regions, the variability of the measured mixing ratios was more affected by chemistry than the other regions indicating that these areas are more remote from the sources and the air had longer chemical processing times indicated by the high b factor ($b_{GA}$ = 0.58 ± 0.38, $b_{MS}$ = 0.48 ± 0.16, $b_{RSS}$ = 0.41 ± 0.21). Similar values have been reported for unpolluted coastal sites (0.56 at Nova

Scotia, Jobson et al., 1998), marine influenced air masses (0.54 at Appledore island, Pszenny et al., 2007; 0.64 at Surinam, Williams et al., 2000) and Eastern Mediterranean Sea (0.49, Arsene et al., 2007) validating our indications for the importance of alkane-radical reactions in these regions.

### 3.3.2 Characterization of radical chemistry

The atmospheric ratio of *i*-butane to *n*-butane has been frequently reported to be between 0.3 and 0.8 for a wide range of

environmental conditions (e.g. Baker et al., 2016; Rossabi and Helmig, 2018). For the AQABA campaign, the isomeric ratio was on average 0.63 ± 0.44 (median = 0.56) with 82.4 % of the samples to between 0.3 and 0.8. The almost identical reaction rate coefficient of the butane isomers (*i*- and *n*-) with the OH radical means that their measured ratio will remain relatively constant and equal to the emission ratio independent of transport times, providing only OH reacts with the species and in-mixing is negligible. Deviation from this range would imply a local source with different emission ratio, or oxidation with

another radical with which the isomers react at different rates. Since the reaction rate coefficient of *n*-butane is about 40 % faster than *i*-butane with Cl, these ratios can be used to investigate the presence of Cl radicals.

As shown in **Fig. 12**, three clusters of data substantially deviate from the slope, displaying different *i*-butane to *n*-butane mixing ratios. The first cluster (termed as case study $CS1_{AG}$, N = 20; black circles) corresponds to the high mixing ratios





encountered in the AG with an isomeric ratio of 1.09 ± 0.18. Considering the small standard deviation of the ratio along with the small geographical coverage, it is more likely that this cluster corresponds to stronger *i*-butane emissions rather than *n*-butane depletion due to reaction with Cl radicals. *I*-butane is more abundant in gas flares (Emam, 2015) which typically contain 45 – 70 % methane (Umukoro and Ismail, 2017). EMF values were on average 35.8 ± 9.5 % indicating that these

particular samples had very high NMHC content, including the gas-flaring product 1-butene which was 3 times higher compared to the AG average. In addition, these samples cover 15 hours, including both day and night times during which Cl concentrations vary substantially. Therefore, Cl reactions are unlikely to be the main reason for the observed anomalous ratio cluster and it is more likely that gas-flaring emissions are responsible for this observation. Another example is the cluster of data that belongs to AS but falls below the commonly observed emission ratio range (CS2$_{AS}$, N = 43; cyan dots). These are

indicative of oceanic emissions of butanes, which are estimated to be emitted at a rate of 0.11 Tg yr$^{-1}$, exceeded in the alkane series only by ethane (0.54 Tg yr$^{-1}$) and propane (0.35 Tg yr$^{-1}$) (Pozzer et al., 2010). Broadgate et al. (1997) reported that *n*-butane emissions from the North Sea are ~4 times stronger that its isomer, possibly explaining the small ratio (0.34 ± 0.1) we observed in the unpolluted region of AS. Finally, the third cluster of measurements (CS3, N = 38, blue squares) were mainly made over the Red Sea in the regions RSS (N = 15) and RSN (N=11). In addition, 12 samples from the GO, MS and SC

exhibit higher *i*- / *n*- butane ratio and were clustered together, in a group than had an average isomeric ratio of 1.84 ± 1.66. The higher and highly variable isomeric butane ratios combined with their comparatively low concentration of both *i*- and *n*-butane, suggest Cl influenced oxidative air masses but nearby ship emissions could also explain this observation.

In the absence of measurement techniques that could quantify Cl radical concentrations directly, the most commonly used method to determine whether Cl chemistry has occurred is the examination of VOC ratios and inter-relationships (Jobson et

al., 1994; Rudolf et al., 1997; Baker et al., 2016). **Fig. 13** shows the correlation between an OH reactive pair (*i*-butane to propane) and a Cl reactive pair (*i*-butane to *n*-butane). If only OH chemistry acts on the air parcel, the *i*-butane to *n*-butane ratio will remain approximately unchanged while the *i*-butane to propane ratio will decrease with processing time since the OH + *i*-butane rate coefficient is twice that of OH + propane (**Table 5**). If only Cl chemistry acts on the air parcel, the opposite is expected: the *i*-butane to propane ratio will remain unchanged while the isomeric ratio will increase with processing since *n*-

butane reacts faster than its isomer with Cl radicals. **Fig. 13** shows that the majority of the samples display an unchanged isomeric butane ratio over two orders of magnitude of *i*-butane to propane ratio, indicating that sources with similar emissions ratios and OH chemistry dominated the air parcels that were encountered along the route. However, a significant portion of the data display isomeric ratios which indicate a mixture of OH and Cl chemical processing.

To evaluate the contribution of OH and Cl radical reactions in the atmospheric processing, kinetic simulations were used to

illustrate the expected tracer ratio evolution (*i*-butane, *n*-butane and propane) starting from average global emission ratios (yellow diamond in **Fig. 13**, Pozzer et al., 2010). The total OH radical concentration was held at 5 x 10$^6$ molecules cm$^{-3}$ and the OH/Cl ratios were adjusted to simulate conditions that range between OH reactions only and Cl reactions only. In order to demonstrate plausible atmospheric ranges for the OH/Cl ratio, the lowest (62:1), highest (108220:1) and average (930:1) ratios were generated in the EMAC model run for the campaign and these have been implemented in **Fig. 13**. In the model the Cl



radical is generated from HCl oxidation; the former being released from sea-spray via acid-displacement (Erickson et al., 1999; Keene et al., 1999; Knipping et al., 2000; Young et al., 2014). Modelled Cl-concentrations are therefore expected to be more abundant over the marine boundary layer and coastal areas. In the atmosphere, $ClNO_2$ represents a further form of reactive chlorine, which is formed in heterogeneous reactions of $N_2O_5$ with sea-salt at night-time and which has been detected

at significant concentrations both in coastal and continental air masses (Thornton et al. 2010, Phillips et al., 2012). $ClNO_2$ can photolyze to produce Cl radicals (Osthoff et al., 2008; Lawler et al., 2009; Riedel et al., 2012; Young et al., 2012) and field observations have shown that $ClNO_2$ can be responsible for more than 50% of near-surface Cl production during early morning (Young et al., 2012; Mielke et al., 2013; Young et al., 2014). As the EMAC model does not include the formation of $ClNO_2$, we may assume that ambient OH/Cl ratios may in fact be lower than the minimum 62:1 ratio generated by the model.

Based on the kinetic model results, the OH/Cl ratio can be expressed as function of the slope (σ) of tracer ratio evolution of these particular hydrocarbons as:

$$\frac{OH}{Cl} = \frac{-80}{\sigma + 0.2} \qquad (4)$$

Where σ corresponds to the slope of ln(*i*-butane/propane) to ln(*i*-butane/*n*-butane) with evolution times illustrated in the colour

scale of **Fig. 13**. **Eq. 4** is independent of the initial emission ratios and of the absolute radical concentrations but it is sensitive to reaction rate coefficients. Therefore, it is only applicable in this form in this region, assuming that the reaction rate coefficients that are presented in **Table 5** are applicable. A more general expression for the relationship of the rate coefficients and OH/Cl ratio under large temperature variations (e.g. free troposphere) was presented in Rudolf et al. (1997).

**Fig. 13** shows that some data points lie on positive slopes which define the unrealistic case in which Cl radicals were the only

oxidant acting on the hydrocarbons. This includes the clusters of points discussed previously (CS1 and CS3), reaffirming that they are unusual emission signatures rather than the result of Cl radical chemistry. It is interesting to note that many of the points in this part of the graph have been affected by our own ship exhaust. Therefore our own ship and possibly other vessels emit propane, *i*-butane and *n*-butane in ratios that are distinctly different from those generated from any combination of OH and Cl radical chemistry acting on the gases when emitted at the average emission ratio (see yellow marker in **Fig. 13**).

Nonetheless, the majority of the samples collected over the RSN and over the AG lie on trend lines that that match the theoretical slope of an OH/Cl ratio ≈ 200:1. As an example, **Fig. 13** illustrates the regression statistics for the majority of the samples collected at RSN (green line, σ = -0.63). Utilizing **Eq. 4**, the resulting radical ratio is 186:1. Our analysis demonstrates that oxidative pairs can be used to provide an indication of the presence of Cl atoms and provide a rough estimate of atmospheric OH to Cl ratios.





## 4. Summary and conclusions

An overview of the atmospheric mixing ratios measured around the Arabian Peninsula during the AQABA ship campaign was provided in terms of regional abundance. Highest mixing ratios were observed over the Arabian Gulf, Suez Canal and the northern parts of Red Sea and the lowest over the Arabian Sea, Gulf of Aden and Mediterranean Sea. The diverse

environmental conditions resulted in wide-ranging (several orders of magnitude) concentrations of NMHCs, the characteristics of which were investigated in relation to regional sources and sinks and air mass ageing/oxidation.

Enhancement ratios between pentane isomers identify the broad O&G activities in Arabian Gulf by displaying a characteristic enhancement of *i*-pentane over *n*-pentane at the order of $0.93 \pm 0.03$ ppb ppb$^{-1}$. Similarly, the enhancement ratio of 1.71 ppb ppb$^{-1}$ that was observed over the Suez Canal and Gulf points towards a characteristic value for ship emissions, found to be

similar with our ship exhaust emission ratios and identical with the ratio slope observed in Texas ship channel (Blake et al., 2014). While the exhaust from the Kommandor Iona could differ in hydrocarbon composition from other vessels, a typical ship exhaust sample in this region was rich in ethene, propane and *n*-butane which consisted ~50% of emitted NMHCs. The highest ethene mixing ratios were observed in the SC ($0.81 \pm 1.11$ ppb), where *n*-butane and propane were highly abundant and correlated with each other. We can therefore conclude that international shipping contributes to atmospheric NMHC

abundance, which should be accounted for in emission inventories. However, atmospheric mixing ratios of ethane were consistently unaffected by marine traffic suggesting that it can be disregarded in emission inventories. Finally, the *i*- / *n*-pentane isomeric ratio characterised a mixture of vehicle emissions and urban pollution in Jeddah port with the value of $2.9 \pm 0.14$ ppb ppb$^{-1}$ to be in line with previously reported observations at this urban centre (Barletta et al., 2017).

Since the atmospheric abundance of propane is influenced by nearby natural gas processing and petroleum refining activities,

its atmospheric mixing ratios were investigated in terms of co-emission with the rest of NMHCs. Besides the ship dominated emissions in the Suez Canal, where propane did not correlate with ethane, higher correlations with other alkanes were observed in the Arabian Gulf and the northern part of the Red Sea. By calculating the excess mole fraction of NMHCs, we showed that the ethane-propane interrelationship is different in hydrocarbon rich associated gas compared to the methane rich non-associated gas which contained higher fractions of propane over ethane, in line with previous observations that

demonstrated the propane dominance in gas leak emissions in the region (Salameh et al., 2016). The atmospheric abundance of ethane and propane over the Arabian Gulf was inconsistent with methane mixing ratios when considering the complete dataset of this region since the different emission ratios are dependent on the source type and characteristics along with the O&G processing method and techniques.

By utilizing the variability-lifetime relationship, we confirmed the extent of remoteness from NMHC sources. The variability

of measurements in the Arabian Gulf and northern Red Sea were mostly determined by the sources. These two regions, together with the Gulf of Oman and Suez Canal showed the smallest dependency on chemical lifetimes. On the other hand, lifetime-variability regression analysis resulted in increased dependencies with the chemical lifetimes over the Mediterranean Sea and Gulf of Aden. We can therefore conclude that these regions are relatively most remote from the NMHC sources.



Interestingly, measurements in the Arabian Sea area, which would have been expected to show the most remote character, showed that the relatively low mixing ratios measured there were from local emissions.

To gain insight into the radical induced oxidation of hydrocarbons around the Arabian Peninsula, we used the *i*-butane/propane and *i*-butane / *n*-butane as oxidative pairs. We found that OH chemistry dominates hydrocarbon removal in the regions of the Middle East sampled in this campaign, although indications for sporadic Cl chemistry were found along the Red Sea and Arabian Gulf.

**Data availability**

The data are archived and distributed through the AQABA repository at the KEEPER service of the Max Planck Digital Library, https://keeper.mpdl.mpg.de and are available to all scientists agreeing to AQABA protocol from August 2019.

**Supplement**

The supplement related to this article is available on-line at https://doi.org/... (link to be specified upon publication).

**Author contributions**

EB performed the measurements, analysed the data and drafted the article. LE performed the measurements. DW analysed the trajectories. JDP is responsible for the methane data, JC for the kinetic calculations and AP for the modelled OH, Cl and $NO_3$ concentrations. JL conceived and realized the project. JW supervised the study. All authors contributed to editing the article.

**Competing interests**

The authors declare no conflict of interest.

**Acknowledgements**

We acknowledge the fruitful collaborations with the Cyprus Institute (CyI), the King Abdullah University of Science and Technology (KAUST) and the Kuwait Institute for Scientific Research (KISR). We are grateful to Hays Ships Ltd, the ship's captain Pavel Kirzner and ship's crew for providing the best possible working conditions on-board Kommandor Iona. We thank all the participants of the AQABA ship campaign and in particular Dr. Harwig Harder for the fruitful discussions and day-to-day organization of the campaign and Dr. Marcel Dorf, Claus Koeppel, Thomas Klüpfel and Rolf Hoffmann for logistics organization and assistance during the setup phase. Finally, we acknowledge Ivan Tadic and Philipp Eger for the use



of a preliminary KI contamination flag. Marc Delmotte, Laurence Vialettes and Olivier Laurent helped with setting up the methane measurements.

## Appendix

### General

| | |
|---|---|
| NMHC | Non Methane Hydrocarbons |
| AQABA | Air Quality and climate in the Arabian Basin |
| KI | Kommandor Iona (ship vessel) |
| O&G | Oil and Gas |

### Regions

| | |
|---|---|
| MS | Mediterranean Sea |
| SC | Suez Canal and Gulf |
| RSN | Red Sea North |
| RSS | Red Sea South |
| GA | Gulf of Aden |
| AS | Arabian Sea |
| GO | Gulf of Oman |
| AG | Arabian Gulf |
| JP | Jeddah Port |

### Methods

| | |
|---|---|
| GC-FID | Gas Chromatography - Flame Ionization Detector |
| VOC | Volatile Organic Compounds |
| HYSPLIT | Hybrid Single Particle Lagrangian Integrated Trajectory model |
| ECHAM5 | 5th generation European Centre Hamburg general circulation model |
| MESSy | Modular Earth Submodel System |
| EMAC | ECHAM/MESSy Atmospheric Chemistry model |
| stp | standard temperature (273.15 K) and pressure (1013.25 hPa) |
| sccm | Standard Cubic Centimeters per Minute (i.e. volume flux for the case of stp) |
| Lpm | Liter per minute at standard temperature and pressure (STP) |
| PTFE | Polytetrafluoroethylene |
| DL | Detection Limit |

### Results

| | |
|---|---|
| ER | enhancement ratio slopes, i.e. the slope term in a linear regression between n-pentane and i-pentane. |
| EMF | Excess Mole Fraction (definition in section 3.2.3) |

**Appendix 1.** Acronyms and abbreviations.



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



**Tables and Figures**

| Compound | DL (ppt) | DL range (ppt) | Total uncertainty (%) |
|---|---|---|---|
| ethane | 7 | 0.1 - 74 | 7.3 |
| propane | 3 | 0.3 - 31 | 7.2 |
| i-butane (2-methylpropane) | 2 | 0.2 - 19 | 8.9 |
| n-butane (butane) | 2 | 0.2 - 22 | 8.5 |
| i-pentane (2-methylbutane) | 2 | 0.2 - 25 | 9.9 |
| n-pentane (pentane) | 2 | 0.2 - 28 | 10.4 |
| i-hexane (2-methylpentane) | 2 | 0.6 - 43 | 6.3 |
| n-hexane (hexane) | 2 | 0.5 - 29 | 7.1 |
| heptane | 1 | 0.5 - 26 | 7.4 |
| octane | 1 | 0.5 - 14 | 7.7 |
| ethene | 25 | 1 - 150 | 12.5 |
| propene | 2 | 0.1 - 29 | 9.5 |
| trans-2-butene (but-2-ene) | 1 | 0.2 - 17 | 11 |
| 1-butene (but-1-ene) | 1 | 0.1 - 12 | 12.4 |
| benzene | 2 | 0.6 - 26 | 7.3 |
| toluene | 2 | 0.5 - 21 | 8.3 |
| m- + p- xylenes (1,3- + 1,4 xylenes) | 3 | 1 - 48 | 8.6 |

**Table 1:** Detection limits and total uncertainty of measured NMHCs. The average detection limits for each compound are reported in the
first column. DL varied during the campaign as result of different sampling volumes and wave height. The average total uncertainty was
derived by error interpolation and is reported in the third column.





| | Area | Mediterranean sea | Suez | Red sea (N) | Red sea (S) | Gulf of Aden | Arabian sea | Gulf of Oman | Arabian Gulf |
|---|---|---|---|---|---|---|---|---|---|
| **Coords (lat.(N), long.(E))** | start | 35.959 , 15.013 | 31.256 , 32.354 | 27.821 , 33.766 | 21.426 , 38.972 | 12.553 , 43.413 | 13.461 , 50.142 | 22.869 , 60.105 | 26.616 , 56.566 |
| | end | 31.256 , 32.354 | 27.821 , 33.766 | 21.426 , 38.972 | 12.553 , 43.413 | 13.461 , 50.142 | 22.869 , 60.105 | 26.616 , 56.566 | 29.381 , 47.953 |
| **ethane** | N | 125 | 90 | 210 | 113 | 116 | 262 | 104 | 162 |
| | mean (ppb) | 0.899 | 2.464 | 3.030 | 0.668 | 0.438 | 0.260 | 0.546 | 7.819 |
| | median (ppb) | 0.878 | 1.575 | 2.215 | 0.540 | 0.427 | 0.236 | 0.420 | 3.745 |
| | SD (ppb) | 0.347 | 2.932 | 2.793 | 0.297 | 0.121 | 0.099 | 0.391 | 9.981 |
| | range (ppb) | 0.506 - 4.173 | 0.267 - 21.587 | 0.457 - 17.329 | 0.454 - 1.973 | 0.274 - 1.108 | 0.155 - 1.336 | 0.229 - 2.433 | 0.403 - 48.02 |
| **propane** | N | 125 | 76 | 189 | 98 | 75 | 178 | 86 | 157 |
| | mean (ppb) | 0.373 | 3.855 | 1.711 | 0.250 | 0.166 | 0.109 | 0.551 | 7.927 |
| | median (ppb) | 0.262 | 1.400 | 1.166 | 0.179 | 0.132 | 0.107 | 0.355 | 2.980 |
| | SD (ppb) | 0.763 | 6.147 | 1.650 | 0.208 | 0.174 | 0.101 | 0.564 | 10.503 |
| | range (ppb) | 0.029 - 8.607 | 0.107 - 32.155 | 0.132 - 10.448 | 0.067 - 1.063 | 0.026 - 1.35 | 0.007 - 1.156 | 0.059 - 2.937 | 0.307 - 53.791 |
| **i- butane** | N | 125 | 76 | 189 | 98 | 72 | 133 | 83 | 158 |
| | mean (ppb) | 0.055 | 1.392 | 0.387 | 0.090 | 0.032 | 0.005 | 0.202 | 2.901 |
| | median (ppb) | 0.028 | 0.501 | 0.245 | 0.048 | 0.008 | 0.003 | 0.127 | 0.552 |
| | SD (ppb) | 0.199 | 2.239 | 0.404 | 0.170 | 0.099 | 0.020 | 0.218 | 4.490 |
| | range (ppb) | 0.003 - 2.216 | 0.015 - 10.644 | 0.017 - 2.398 | 0.007 - 1.249 | 0.003 - 0.621 | 0.001 - 0.230 | 0.005 - 1.067 | 0.052 - 14.733 |
| **n- butane** | N | 123 | 76 | 189 | 97 | 73 | 159 | 83 | 158 |
| | mean (ppb) | 0.109 | 3.009 | 0.749 | 0.090 | 0.043 | 0.009 | 0.368 | 4.193 |
| | median (ppb) | 0.045 | 0.984 | 0.434 | 0.043 | 0.013 | 0.004 | 0.244 | 1.145 |
| | SD (ppb) | 0.555 | 5.085 | 0.848 | 0.170 | 0.138 | 0.048 | 0.435 | 5.568 |
| | range (ppb) | 0.008 - 6.180 | 0.027 - 27.701 | 0.023 - 5.152 | 0.008 - 1.517 | 0.003 - 1.074 | 0.001 - 0.608 | 0.014 - 2.217 | 0.089 - 26.875 |
| **i- pentane** | N | 116 | 76 | 188 | 91 | 64 | 79 | 85 | 158 |
| | mean (ppb) | 0.052 | 0.802 | 0.297 | 0.099 | 0.044 | 0.006 | 0.271 | 1.499 |
| | median (ppb) | 0.019 | 0.349 | 0.154 | 0.044 | 0.017 | 0.001 | 0.145 | 0.508 |
| | SD (ppb) | 0.236 | 1.088 | 0.379 | 0.333 | 0.086 | 0.020 | 0.323 | 1.987 |
| | range (ppb) | 0.003 - 2.447 | 0.011 - 5.337 | 0.005 - 2.119 | 0.002 - 3.141 | 0.001 - 0.510 | 0.001 - 0.178 | 0.009 1.545 | 0.018 - 11.420 |
| **n- pentane** | N | 108 | 76 | 188 | 96 | 74 | 165 | 84 | 158 |
| | mean (ppb) | 0.043 | 0.481 | 0.230 | 0.055 | 0.033 | 0.005 | 0.159 | 1.472 |
| | median (ppb) | 0.014 | 0.232 | 0.125 | 0.037 | 0.009 | 0.004 | 0.093 | 0.329 |
| | SD (ppb) | 0.254 | 0.632 | 0.301 | 0.104 | 0.089 | 0.015 | 0.180 | 2.109 |
| | range (ppb) | 0.002 - 2.643 | 0.007 - 2.885 | 0.002 - 1.777 | 0.003 - 0.934 | 0.002 - 0.561 | 0.002 - 0.195 | 0.007 - 1.022 | 0.014 - 12.033 |
| **i-hexane** | N | 21 | 46 | 165 | 49 | 19 | 13 | 57 | 73 |
| | mean (ppb) | 0.058 | 0.222 | 0.096 | 0.033 | 0.032 | 0.032 | 0.062 | 0.126 |
| | median (ppb) | 0.010 | 0.116 | 0.047 | 0.021 | 0.020 | 0.007 | 0.038 | 0.059 |
| | SD (ppb) | 0.187 | 0.227 | 0.118 | 0.061 | 0.036 | 0.048 | 0.064 | 0.173 |
| | range (ppb) | 0.002 - 0.861 | 0.003 - 0.775 | 0.008 - 0.532 | 0.004 - 0.422 | 0.006 - 0.138 | 0.004 - 0.135 | 0.005 - 0.331 | 0.004 - 0.967 |
| **n-hexane** | N | 56 | 60 | 169 | 49 | 28 | 6 | 59 | 69 |
| | mean (ppb) | 0.035 | 0.141 | 0.079 | 0.025 | 0.023 | 0.034 | 0.063 | 0.120 |
| | median (ppb) | 0.019 | 0.062 | 0.047 | 0.015 | 0.011 | 0.041 | 0.034 | 0.060 |
| | SD (ppb) | 0.120 | 0.158 | 0.093 | 0.034 | 0.044 | 0.019 | 0.085 | 0.195 |
| | range (ppb) | 0.001 - 0.915 | 0.005 - 0.589 | 0.002 - 0.426 | 0.001 - 0.193 | 0.002 - 0.228 | 0.003 - 0.05 | 0.007 - 0.481 | 0.009 - 1.375 |
| **n-heptane** | N | 9 | 51 | 141 | 46 | 21 | 11 | 50 | 59 |
| | mean (ppb) | 0.043 | 0.086 | 0.031 | 0.010 | 0.013 | 0.016 | 0.021 | 0.030 |
| | median (ppb) | 0.010 | 0.038 | 0.013 | 0.008 | 0.005 | 0.007 | 0.011 | 0.016 |
| | SD (ppb) | 0.101 | 0.099 | 0.043 | 0.007 | 0.027 | 0.021 | 0.026 | 0.056 |
| | range (ppb) | 0.002 - 0.313 | 0.002 - 0.335 | 0.001 - 0.179 | 0.002 - 0.038 | 0.001 - 0.118 | 0.001 - 0.058 | 0.001 - 0.141 | 0.002 - 0.392 |
| **octane** | N | 20 | 54 | 141 | 35 | 20 | 4 | 47 | 68 |
| | mean (ppb) | 0.008 | 0.044 | 0.016 | 0.008 | 0.008 | 0.062 | 0.010 | 0.011 |
| | median (ppb) | 0.002 | 0.018 | 0.008 | 0.005 | 0.003 | 0.063 | 0.006 | 0.007 |
| | SD (ppb) | 0.022 | 0.055 | 0.018 | 0.011 | 0.014 | 0.066 | 0.013 | 0.013 |
| | range (ppb) | 0.001 - 0.1 | 0.002 - 0.193 | 0.001 - 0.075 | 0.001 - 0.049 | 0.001 - 0.06 | 0.005 - 0.120 | 0.002 - 0.067 | 0.001 - 0.094 |
| **ethene** | N | 14 | 57 | 130 | 54 | 51 | 104 | 45 | 137 |
| | mean (ppb) | 0.114 | 0.809 | 0.136 | 0.108 | 0.063 | 0.090 | 0.391 | 0.634 |
| | median (ppb) | 0.107 | 0.340 | 0.110 | 0.040 | 0.055 | 0.084 | 0.068 | 0.206 |
| | SD (ppb) | 0.043 | 1.110 | 0.111 | 0.317 | 0.042 | 0.057 | 1.975 | 0.860 |
| | range (ppb) | 0.037 - 0.204 | 0.003 - 5.594 | 0.001 - 0.640 | 0.006 - 2.238 | 0.016 - 0.258 | 0.004 - 0.240 | 0.005 - 13.332 | 0.001 - 3.581 |
| **propene** | N | 104 | 76 | 189 | 92 | 73 | 175 | 83 | 158 |
| | mean (ppb) | 0.012 | 0.152 | 0.016 | 0.022 | 0.020 | 0.012 | 0.041 | 0.075 |
| | median (ppb) | 0.010 | 0.043 | 0.015 | 0.011 | 0.012 | 0.011 | 0.014 | 0.022 |
| | SD (ppb) | 0.008 | 0.272 | 0.009 | 0.065 | 0.059 | 0.008 | 0.184 | 0.113 |
| | range (ppb) | 0.001 - 0.06 | 0.009 - 1.932 | 0.003 - 0.061 | 0.005 - 0.560 | 0.001 - 0.516 | 0.002 - 0.074 | 0.004 - 1.684 | 0.001 - 0.556 |
| **trans-2- butene** | N | 1 | 33 | 9 | 33 | 36 | 175 | 69 | 148 |
| | mean (ppb) | n.a. | 0.010 | 0.008 | 0.007 | 0.007 | 0.006 | 0.008 | 0.018 |
| | median (ppb) | n.a. | 0.005 | 0.008 | 0.006 | 0.007 | 0.006 | 0.007 | 0.010 |
| | SD (ppb) | n.a. | 0.017 | 0.006 | 0.005 | 0.001 | 0.002 | 0.006 | 0.016 |
| | range (ppb) | 0.002 | 0.001 - 0.103 | 0.002 - 0.019 | 0.001 - 0.026 | 0.005 - 0.009 | 0.002 - 0.012 | 0.005 - 0.053 | 0.004 - 0.077 |
| **1-butene** | N | 16 | 62 | 44 | 48 | 40 | 89 | 44 | 108 |
| | mean (ppb) | 0.004 | 0.034 | 0.007 | 0.007 | 0.005 | 0.004 | 0.014 | 0.025 |
| | median (ppb) | 0.002 | 0.013 | 0.005 | 0.004 | 0.004 | 0.004 | 0.005 | 0.009 |
| | SD (ppb) | 0.005 | 0.052 | 0.004 | 0.019 | 0.002 | 0.002 | 0.054 | 0.030 |
| | range (ppb) | 0.001 - 0.019 | 0.002 - 0.349 | 0.002 - 0.018 | 0.001 - 0.121 | 0.002 - 0.012 | 0.002 - 0.025 | 0.001 - 0.361 | 0.002 - 0.125 |
| **benzene** | N | 109 | 60 | 165 | 77 | 74 | 152 | 42 | 77 |
| | mean (ppb) | 0.057 | 0.160 | 0.069 | 0.034 | 0.026 | 0.021 | 0.078 | 0.121 |
| | median (ppb) | 0.052 | 0.099 | 0.061 | 0.030 | 0.023 | 0.015 | 0.065 | 0.095 |
| | SD (ppb) | 0.022 | 0.183 | 0.036 | 0.021 | 0.013 | 0.029 | 0.079 | 0.072 |
| | range (ppb) | 0.028 - 0.170 | 0.027 - 0.734 | 0.025 - 0.196 | 0.011 - 0.191 | 0.011 - 0.079 | 0.007 - 0.255 | 0.024 - 0.546 | 0.059 - 0.391 |
| **toluene** | N | 109 | 60 | 172 | 76 | 74 | 124 | 68 | 77 |
| | mean (ppb) | 0.022 | 0.276 | 0.045 | 0.047 | 0.019 | 0.012 | 0.043 | 0.046 |
| | median (ppb) | 0.017 | 0.088 | 0.028 | 0.021 | 0.008 | 0.009 | 0.022 | 0.031 |
| | SD (ppb) | 0.022 | 0.447 | 0.053 | 0.056 | 0.028 | 0.022 | 0.058 | 0.048 |
| | range (ppb) | 0.003 - 0.2 | 0.008 - 1.686 | 0.003 - 0.34 | 0.002 - 0.361 | 0.004 - 0.203 | 0.001 - 0.178 | 0.005 - 0.281 | 0.001 - 0.322 |
| **m-, p- xylenes** | N | 92 | 60 | 171 | 72 | 64 | 56 | 42 | 68 |
| | mean (ppb) | 0.015 | 0.107 | 0.043 | 0.046 | 0.020 | 0.020 | 0.098 | 0.087 |
| | median (ppb) | 0.011 | 0.036 | 0.026 | 0.031 | 0.015 | 0.016 | 0.041 | 0.062 |
| | SD (ppb) | 0.016 | 0.164 | 0.054 | 0.044 | 0.017 | 0.012 | 0.101 | 0.083 |
| | range (ppb) | 0.002 - 0.102 | 0.003 - 0.606 | 0.005 - 0.359 | 0.005 - 0.186 | 0.003 - 0.112 | 0.006 - 0.083 | 0.012 - 0.403 | 0.008 - 0.504 |

**Table 2.** Spatial volume mixing ratio means, medians, standard deviations (SD) and ranges of measured NMHCs.



| | SC | | | | RSN | | | | AG | | | |
|---|---|---|---|---|---|---|---|---|---|---|---|---|
| | $ER_{VOC/C3H8}$ (ppb ppb$^{-1}$) | $R^2$ | $ER_{VOC/CH4}$ (ppt ppb$^{-1}$) | $R^2$ | $ER_{VOC/C3H8}$ (ppb ppb$^{-1}$) | $R^2$ | $ER_{VOC/CH4}$ (ppt ppb$^{-1}$) | $R^2$ | $ER_{VOC/C3H8}$ (ppb ppb$^{-1}$) | $R^2$ | $ER_{VOC/CH4}$ (ppt ppb$^{-1}$) | $R^2$ |
| ethane | 0.35 | 0.22 | 6.8 | 0.57 | **1.3** | **0.95** | 64 | 0.67 | **0.91** | **0.9** | 106.5 | 0.23 |
| propane | | | **18.4** | **0.71** | | | **47.8** | **0.7** | | | 115 | 0.35 |
| i-butane | **0.43** | **0.89** | 9.7 | 0.91 | 0.22 | 0.94 | **11.8** | **0.75** | 0.39 | 0.83 | 45.1 | 0.3 |
| n-butane | **1.1** | **0.91** | 20.1 | 0.82 | **0.5** | **0.96** | 24.5 | 0.7 | **0.52** | **0.97** | 63.6 | 0.38 |
| i-pentane | 0.19 | 0.51 | 4.9 | 0.63 | **0.22** | **0.91** | 10.2 | 0.62 | **0.18** | **0.92** | 21.4 | 0.34 |
| n-pentane | 0.11 | 0.5 | 2.5 | 0.57 | **0.17** | **0.91** | 8 | 0.59 | **0.19** | **0.92** | 22.8 | 0.34 |
| benzene | **0.1** | **0.91** | **1.5** | **0.83** | 0.017 | 0.47 | **1.1** | **0.72** | 0.019 | 0.3 | 0.89 | 0.12 |
| toluene | **0.24** | **0.87** | 2.4 | 0.11 | 0.025 | 0.59 | 1.2 | 0.53 | 0.0005 | 0.01 | 0.29 | 0.03 |
| xylenes | **0.09** | **0.87** | 0.9 | 0.12 | 0.001 | 0.01 | 0.27 | 0.03 | 0.0012 | 0.01 | 0.1 | 0.12 |

**Table 3**. Correlation matrix of enhancement volume mixing ratio slopes for selected alkanes and aromatics, using propane and

5   methane as tracer compounds.



| Area | [OH] (molecules cm$^{-3}$) | [Cl] (molecules cm$^{-3}$) | [NO$_3$] (molecules cm$^{-3}$) |
|---|---|---|---|
| MS | 2.4 x 10$^6$ | 6.8 x 10$^3$ | 1.1 x 10$^8$ |
| SC | 5.2 x 10$^6$ | 1.1 x 10$^4$ | 3.3 x 10$^8$ |
| RSN | 4.4 x 10$^6$ | 1.9 x 10$^3$ | 5.3 x 10$^8$ |
| RSS | 4.5 x 10$^6$ | 0.5 x 10$^3$ | 4.7 x 10$^8$ |
| GA | 3.3 x 10$^6$ | 9.9 x 10$^3$ | 1.6 x 10$^8$ |
| AS | 2.3 x 10$^6$ | 1.3 x 10$^4$ | 0.9 x 10$^7$ |
| GO | 5.3 x 10$^6$ | 1.9 x 10$^3$ | 5.2 x 10$^8$ |
| AG | 4.6 x 10$^6$ | 2.3 x 10$^3$ | 9.8 x 10$^8$ |

**Table 4.** Spatial average OH and Cl concentrations simulated by MOM.

| Compound | k$_{OH}$ (cm$^3$ molecule$^{-1}$ s$^{-1}$) | k$_{CL}$ (cm$^3$ molecule$^{-1}$ s$^{-1}$) | k$_{NO3}$ (cm$^3$ molecule$^{-1}$ s$^{-1}$) |
|---|---|---|---|
| ethane | 2.5 x 10$^{-13}$ | 5.9 x 10$^{-11}$ | 1.0 x 10$^{-17}$ |
| propane | 1.1 x 10$^{-13}$ | 1.4 x 10$^{-10}$ | 7.0 x 10$^{-17}$ |
| i-butane | 2.1 x 10$^{-13}$ | 1.4 x 10$^{-10}$ | 1.1 x 10$^{-16}$ |
| n-butane | 2.3 x 10$^{-13}$ | 2.2 x 10$^{-10}$ | 4.6 x 10$^{-17}$ |
| i-pentane | 3.6 x 10$^{-13}$ | 2.2 x 10$^{-10}$ | 1.6 x 10$^{-16}$ |
| n-pentane | 3.8 x 10$^{-13}$ | 2.8 x 10$^{-10}$ | 8.7 x 10$^{-17}$ |

**Table 5.** Reaction rate constants of C2-C5 alkanes.





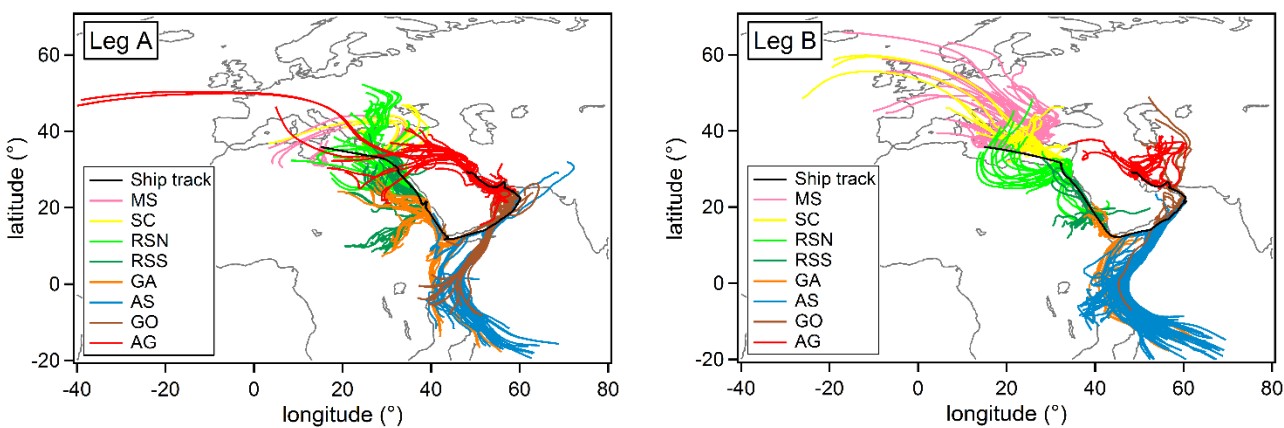

**Figure 1.** Back-trajectory calculations along the route which start at the ship position and follow air masses back in time for four days on an hourly time grid.





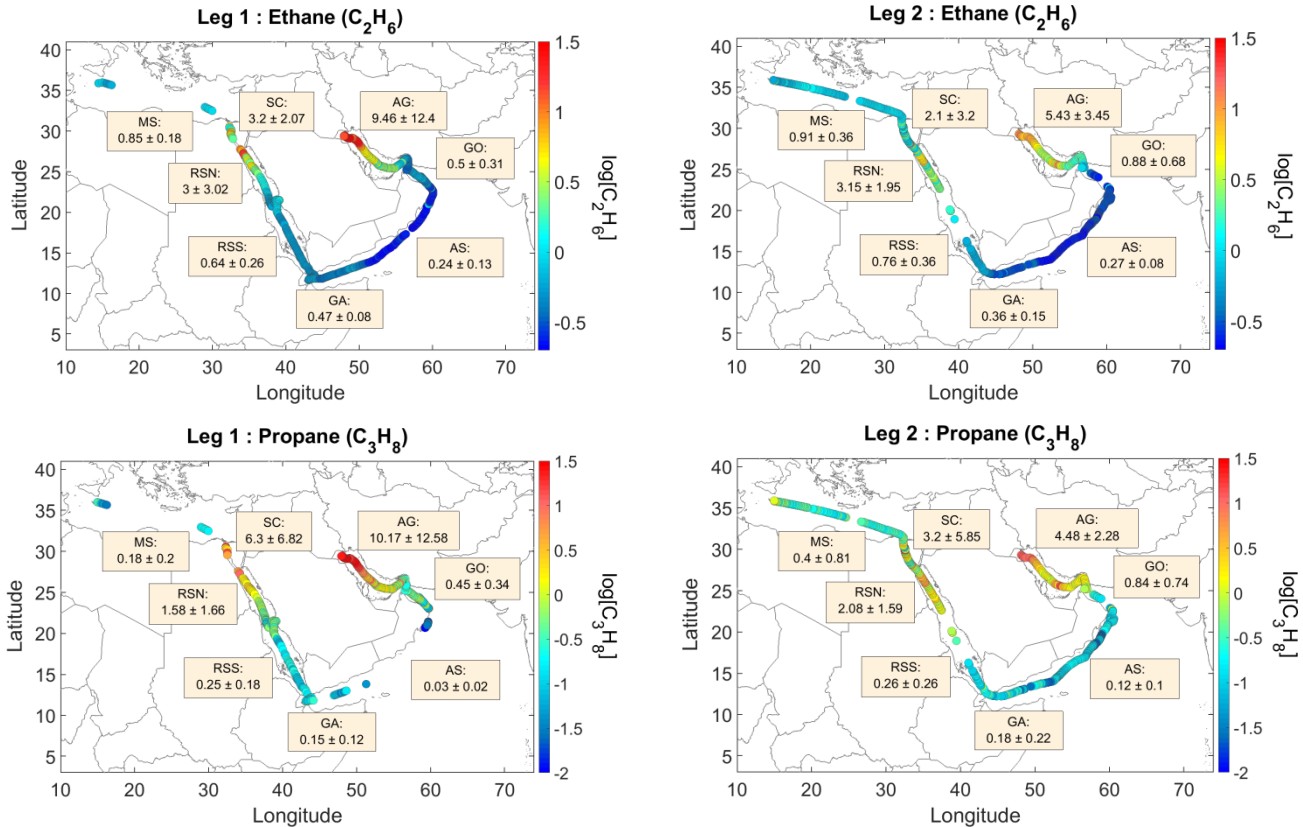

Figure 2. Ethane and propane volume mixing ratios during AQABA. Average mixing ratios ± standard deviation (in ppb) for each area and leg. More detailed statistics are provided in the supplement (Fig. S3 - S4) and in Table 2.



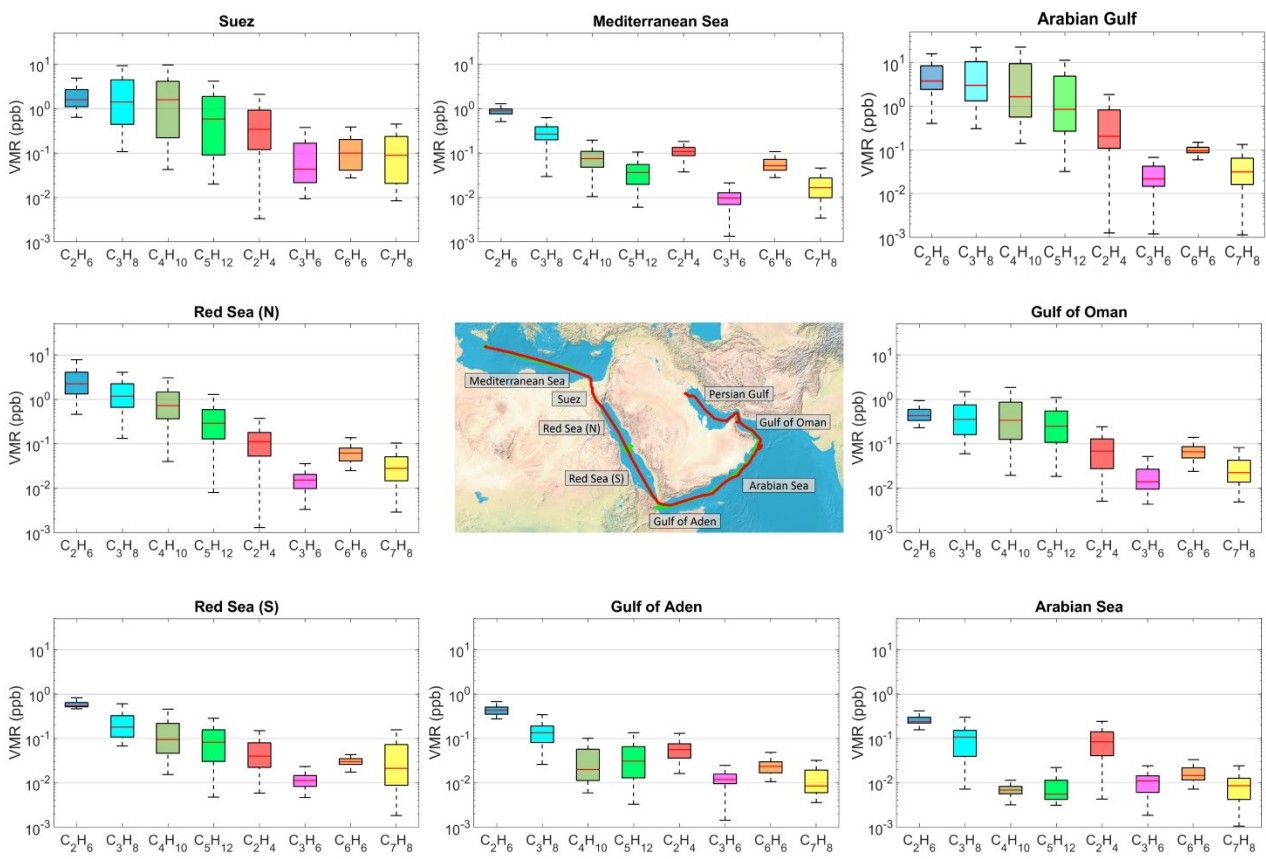

**Figure 3.** Volume mixing ratios of selected NMHC species over the eight regions. For each box, the central red line indicates the median mixing ratio for both campaign legs. The bottom and the top edges of the box indicate the 25th (q1) and 75th (q3) percentile respectively. The boxplot draws points as outliers if they are greater than q3 + w × (q3 − q1) or less than q1 − w × (q3 − q1). The whiskers correspond to ± 2.7σ and 99.3 % coverage if the data are normally distributed. The ship track of the first leg is shown in the map with the green line, the second leg with the red line.





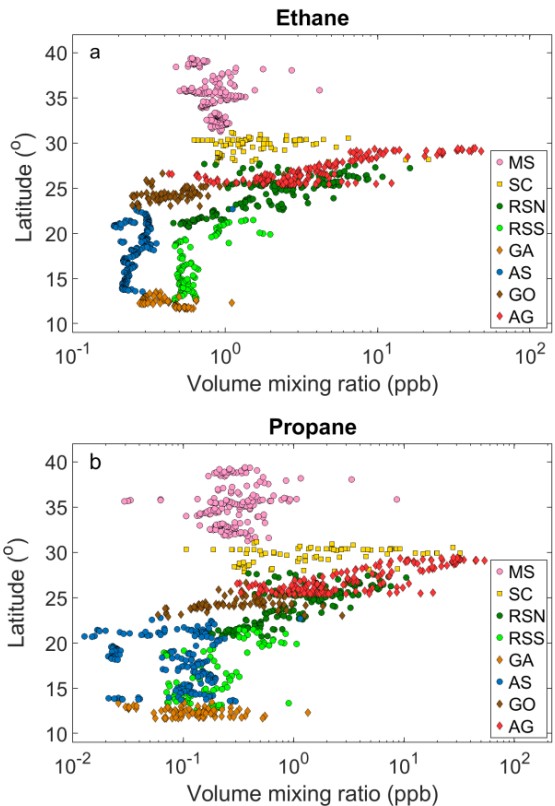

**Figure 4.** Latitudinal distribution of ethane (a) and propane (b) volume mixing ratios.



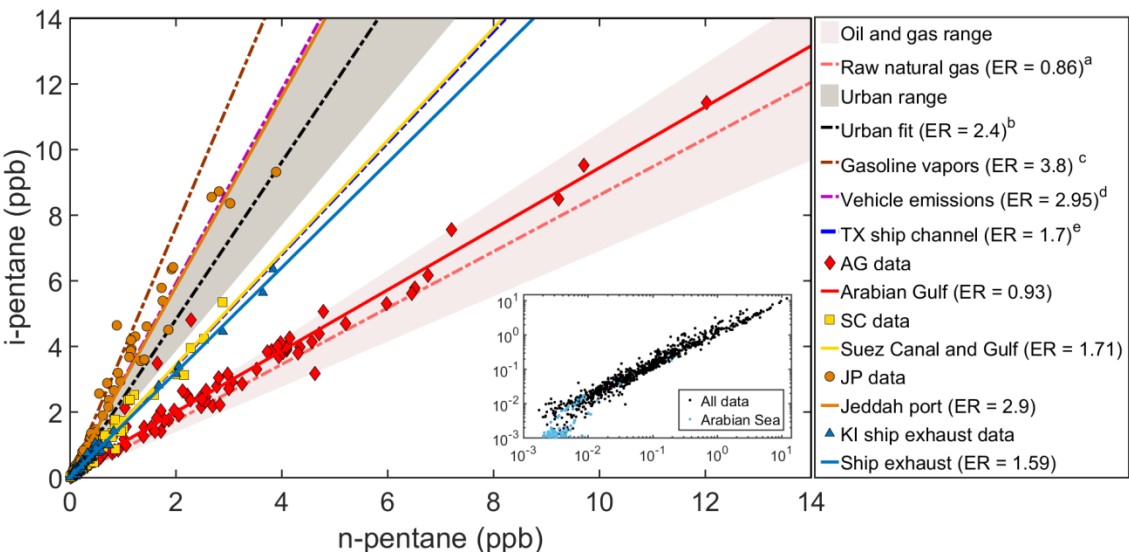

**Figure 5.** Relationship between pentane isomers. Linear fits of the data obtained in the Arabian Gulf, Suez Canal, Jeddah port are compared to enhancement ratio slopes (ER) from the literature with values for raw and natural gas ([a]Gilman et al., 2013), urban areas ([b]Baker et al. 2008), gasoline vapors ([c]Gentner et al., 2009), vehicle emissions ([d]Broderick and Marnane, 2002) and Texas ship channel ([e]Blake et al., 2014). Complete regression statistics: AG: 0.93x + 0.13, $R^2$ = 0.97, SC: 1.71x – 0.02, $R^2$ = 0.98, JP: 2.9x + 0.09, $R^2$ = 0.96, KI ship exhaust: 1.59x + 0.01, $R^2$ = 0.99.



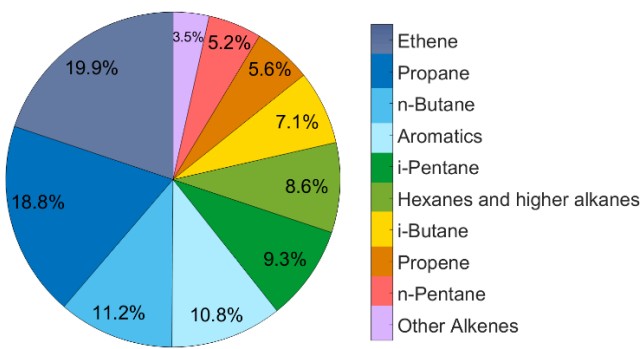

**Figure 6.** Composition of a typical KI ship exhaust sample. Aromatics include benzene (2.8 %), toluene (4.9 %) and *m*-, *p*-xylenes (3.1 %), hexanes and higher alkanes include *i*-hexane (2.6 %), *n*-hexane (2.7 %), *n*-heptane (2.2 %) and octane (1.2 %), and other alkanes include trans-2-butene (0.8 %), 1-butene (1.5 %) and 1-pentene (1.2 %). The sample was measured over the Arabian Sea and background ambient mixing ratios were subtracted prior to relative composition calculation. Ethane mixing ratios were 28 ppt lower inside the exhaust sample, being within the uncertainty range of the measurement.



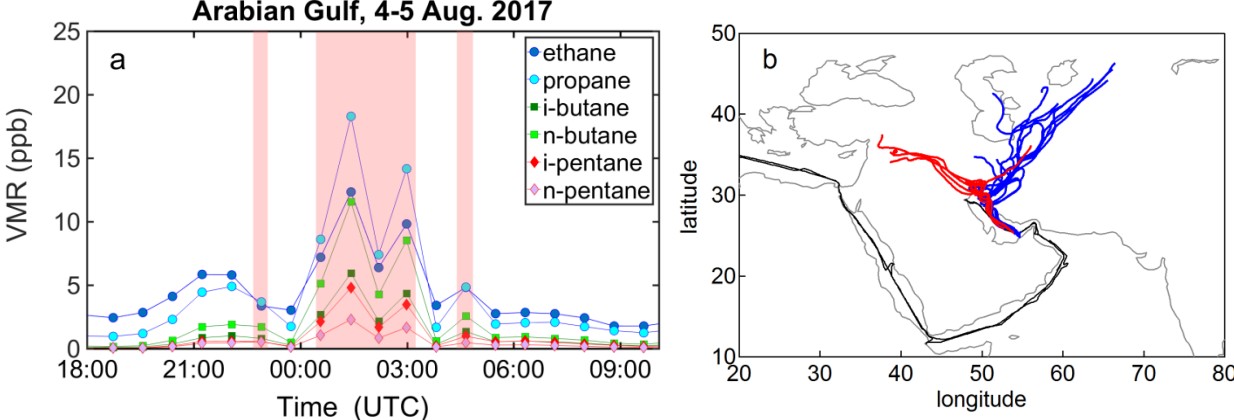

**Figure 7.** Time series of C2-C5 alkane volume mixing ratios (a) and back-trajectories (b) of samples collected in AG on the 4-5 August 2017. In (b) the red trajectories correspond to the shaded red data points of (a) while the blue trajectories correspond to all other samples. The back-trajectories start at the ship position and follow the air masses back in time for six days on an hourly time grid.





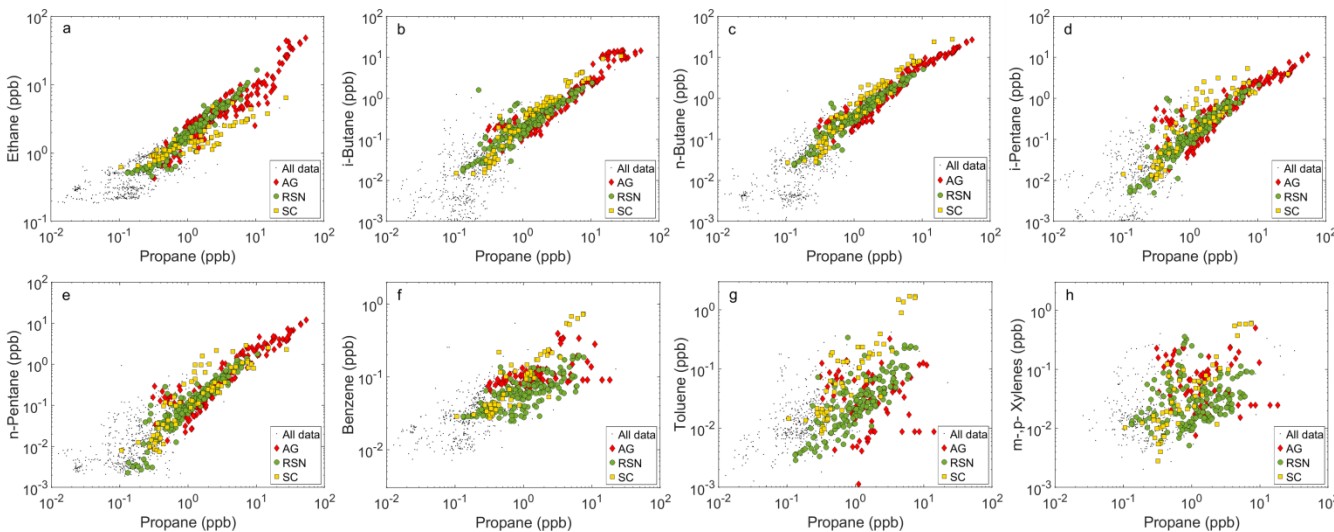

**Figure 8.** Correlations between propane and selected NMHCs for the areas of Arabian Gulf (AG, red diamonds), Red Sea North (RSN, green circles) and Suez Canal and Gulf (SC, yellow squares).





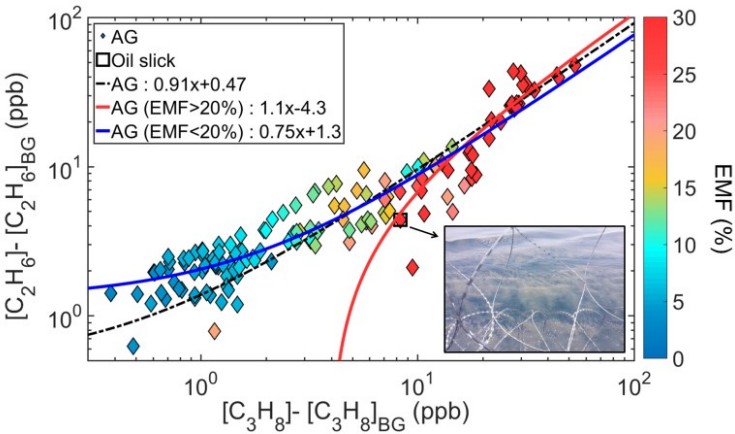

**Figure 9.** Excess mixing ratios of ethane and propane coloured by the excess NMHC mole fraction (EMF) of each sample for the area of Arabian Gulf (AG).



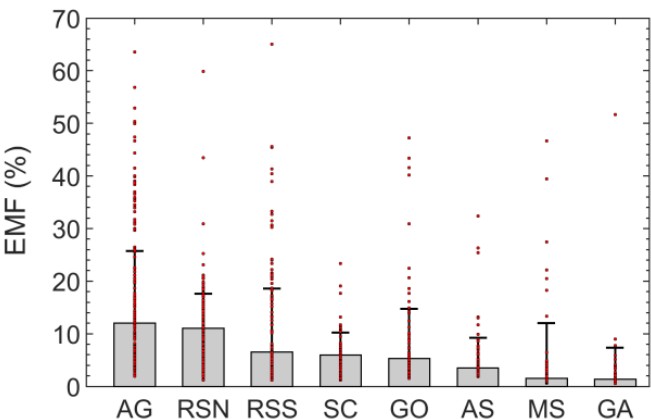

**Figure 10.** Spatial excess NMHC mole fractions. Bars represent the regional median and error bars the standard deviation. Individual samples are shown with red dots.





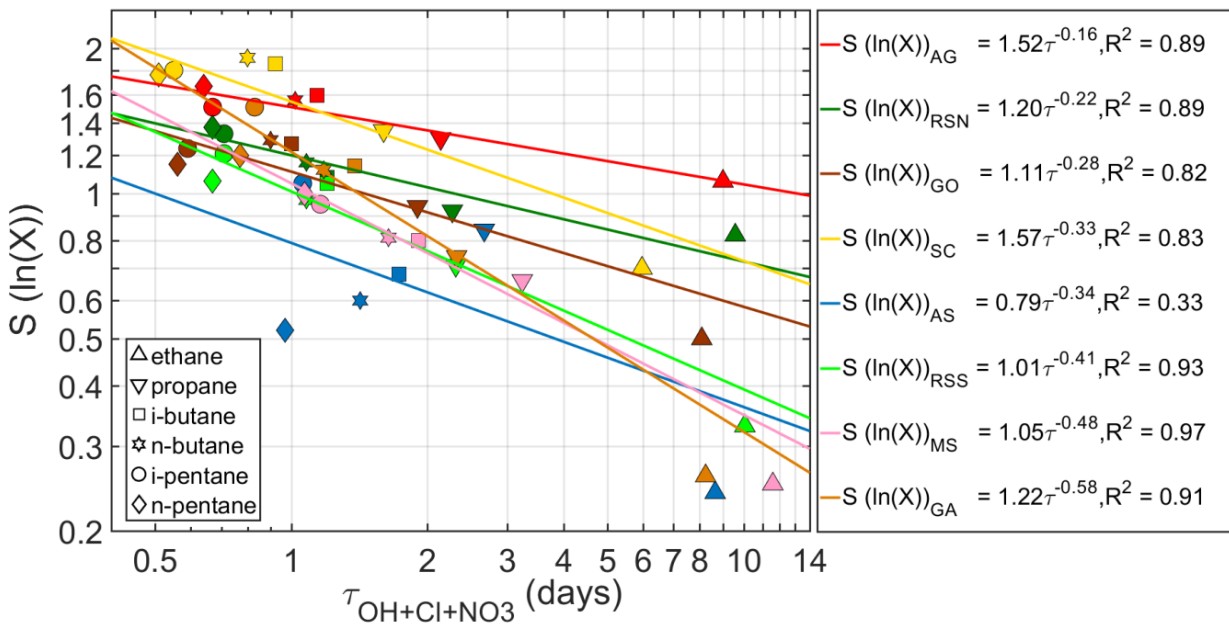

**Figure 11.** Lifetime-variability relationship for C2-C5 alkanes in all regions.



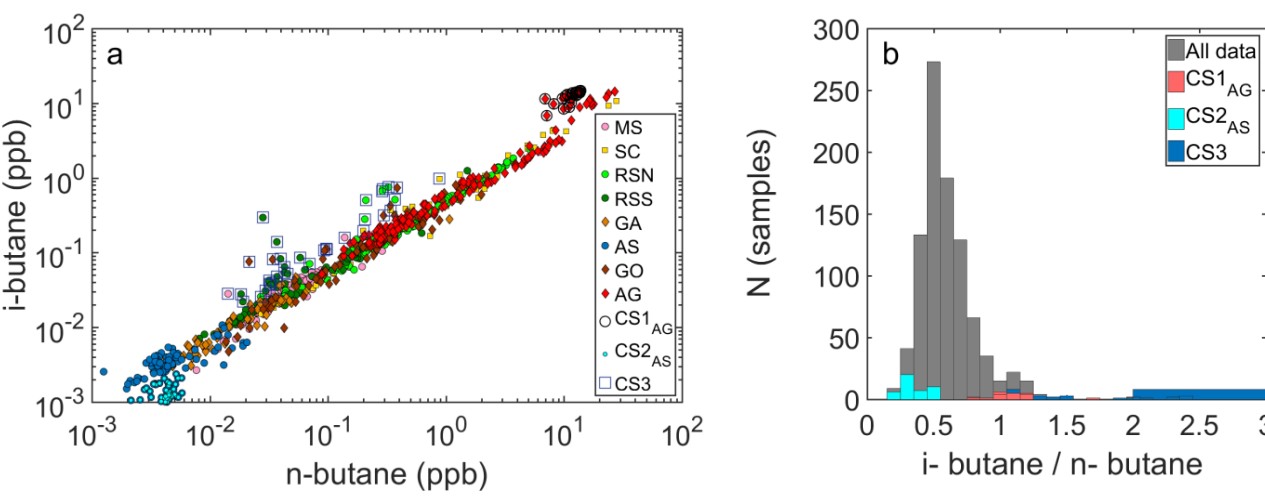

**Figure 12.** (a) Relationship between pentane isomers and (b) distribution of the *i*-butane / *n*-butane ratio considering all data (grey bars) and the case study data (blue bars) as they have been selected from (a).

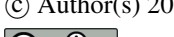



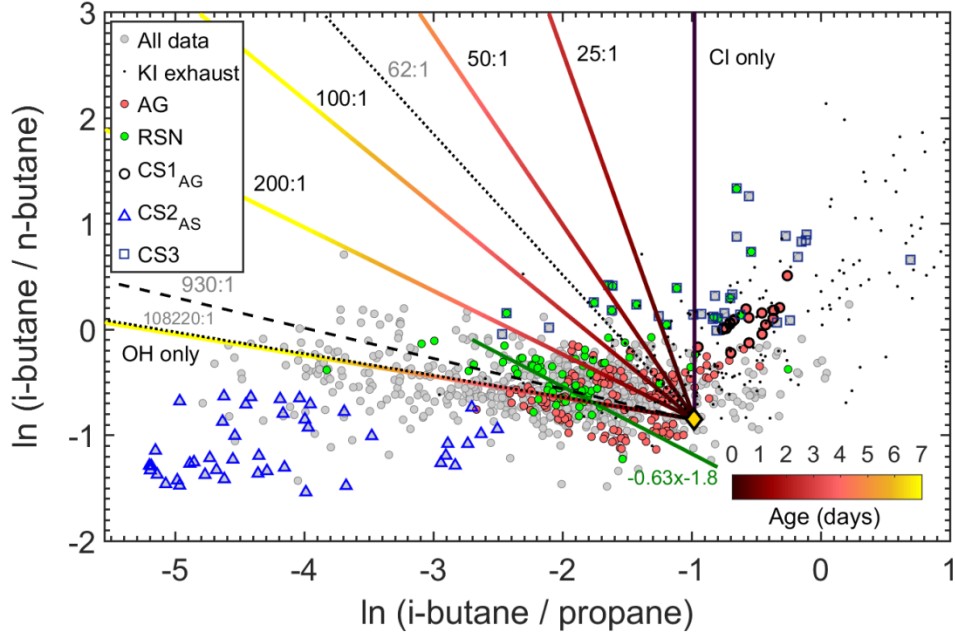

**Figure 13.** Observations of selected VOC tracer ratios (*i*-butane, *n*-butane, propane) together with their expected evolution for OH and Cl oxidation from global emission ratios (yellow diamond). The dashed lines indicate the lowest (62:1), highest (108220:1) and average (930:1) OH to Cl ratios derived from the EMAC model.