# Peer review of "Non Methane Hydrocarbon (C2-C8) sources and sinks around the Arabian Peninsula"

_Atmospheric Chemistry and Physics, 2019_

## Referee Comment (RC1) · Anonymous Referee #1 · 21 Mar 2019

The authors measured speciated hydrocarbon concentrations from a research vessel that traveled around across the Mediterranean Sea, through the Suez Canal, and around the Arabian Peninsula. This region is both data-scarce and source-rich, especially in the Persian Gulf where there are extensive Oil & Gas (O&G) activities. Analyses of the data presented in the manuscript lead to interesting findings about sources that contribute to observed hydrocarbon levels, as well as the importance of various chemical removal processes such as reactions with the hydroxyl and chlorine radicals. I recommend the paper be accepted after minor revisions are made to address the following comments.

1. The authors need to consider more carefully the precision of measured data and calculated ratios. It is not reasonable to report measured values with 4 or 5 digits of precision given what the authors have written about uncertainties in the measurement methods (see first paragraph on page 6). Based on what the authors have written there, I would say two digits of precision could be readily justified, but no way should values be reported with four or five significant figures.

Table 2 requires rethinking and extensive revisions to report all of the tabulated results with defensible precision. This issue also arises on page 9: 13.33 ppb of ethene at line 6 and similarly in numerous instances between lines 11 and 19 on the same page. On page 10: I would write 0.81, 0.96 and 0.88 for the ERs, and at line 16, I would write 27 and 54 ppb. See also page 15, line 4, and page 16, line 33 (108220:1, which appears again on line 5 of page 41 in the last figure caption.

2. While some Arab governments prefer to use the term "Arabian Gulf", the main body of water between Iran and Saudi Arabia is still known internationally as the Persian Gulf. The authors should either use "Persian Gulf" or at least note the equivalence between Arabian Gulf and Persian Gulf in the text to reduce potential for confusion among readers. On page 7, line 20, Suez Gulf is typically referred to as the Gulf of Suez.

3. On page 9, line 5, the high ethene levels are linked to O&G activities, but to my knowledge O&G activities do not emit olefins. There can be a petrochemical source of ethene (e.g., in Houston) for example when ethane is converted to ethene and subsequently polymerized to make polyethylene. Also ethene is present in vehicle exhaust emissions.

4. Page 11, line 25, do you mean O&G instead of O&N here?

5. Page 13, lines 13 and 33, do you mean excess mole fraction? The meaning of "the mole fraction of this sample" is unclear.

---

## Referee Comment (RC2) · Anonymous Referee #2 · 11 Apr 2019

This study reports the observed mixing ratios and relevant enhancement ratios for simple hydrocarbons including the light alkanes, alkenes, and C6-C8 aromatics via shipborne measurements conducted around the entire Arabian Peninsula on two separate transects. The authors collected high-quality data in an understudied region that contains many diverse emission sources including shipping, urban, oil and natural gas operations as well as very clean conditions with signatures of biogenic marine sources. Each of these sources was carefully identified using literature values and back trajectories to help determine likely source regions. The authors investigated how various emission source profiles and/or atmospheric oxidation would impact the observed enhancement ratios and variability in observed mixing ratios.

[Figure]

This was a well-written manuscript. I recommend publication after a few small changes/clarifications.

Technical comments in order of appearance:

P1, L14 – Specify "This region. . ." as the Middle East, Arabian Gulf, etc., which ever term is most appropriate.

P3, L30 – ". . .uses a dual-stage pre-concentration principle, additionally equipped with a focusing trap and a stripper column. . ." It was unclear to me if the focusing trap and the stripper column were part of the "pre-concentration principle" or occur after the sample pre-concentration. What exactly does the stripper column do? Does it remove the permanent gases, the lightest alkanes, etc.? A little more information would make this clear to the readers.

P4, L17 – I would state the differences as factors as it is easier to quickly comprehend (is 120% = a factor of 2.2 or 1.2?). Be sure to include the sign of the "difference," i.e.., positive or negative artifacts.

P7, L22 – Please clarify what "this geographical demarcation" means. Do you mean "each region"? Perhaps, consider combining Figures 1 and 2 for ease of comparison with this statement.

P8, L1 – Curious! Did you observe a diurnal profile in ethene in this region, increasing with daylight hours? The marine boundary layer height usually doesn't change much over open water, so a diurnal profile (particularly higher daytime concentrations) would bolster your hypothesis of a photo-sensitive marine biogenic source.

P8, L7 – "Interestingly" appears five times and starts to become a bit redundant after the second time. Please consider using sparingly, especially since it is a subjective term.

P8, L14 – Why would you "account for both isomers" when referring to the butanes? I'm not sure what is gained from that.

P8, L29 – Figure S20 does not show that the benzene to toluene ratio only the distribution of the mixing ratios. You could easily show this in a separate graph. Please include references or how you came to the conclusion that benzene/toluene > 1 indicates fresh ship emissions and/or biomass burning.

P8, L32-33 – Please list what the "marine traffic associated gases" are or refer to Figure 6 to be discussed in section 3.2.1.

P13, L12 – A pie chart similar to the ship emissions detailing the composition of this sample would be great to include, even in the SI.

P13, L13-14 – The authors state that it is a crude oil slick (L11), but are now calling it "associated gas." Also, beware of the "weathering effect" of oil slicks as the most volatile species tend to evaporate first and do not necessarily represent the actual composition of the starting material.

P31 –Include the region names as you did with the previous table. Also, are propane and methane really tracer compounds? If so, of what? Best to keep it simple and just state that you are presenting the VOC to propane or methane observed enhancement ratios.

P38 – Ethane was lower inside the ship plume than the background values?

P41 – Include what the picture is of and any photo credits. I'm assuming it's of THE oil slick, not just a random oil slick, but I'm confused as it looks like barbed wire is in the foreground(?). This makes me think that the picture was not taken from the ship. Perhaps best to simply leave out.

P44 – Check the caption. Replace pentane with butane. There are also red bars for case study #1.

---

## Author Comment (AC1) · 25 Apr 2019

**Non Methane Hydrocarbon (C2–C8) sources and sinks around the Arabian Peninsula (acp-2019-92): Final response**

We would like to thank the referees for recognizing the significance of our work and for providing valuable feedback that corrects some minor misconceptions in the ACPD version. Below, we answer in detail to each point raised and demonstrate the changes that have been implemented in the revised version.

**Anonymous Referee #1**

**General evaluation**: The authors measured speciated hydrocarbon concentrations from a research vessel that traveled around across the Mediterranean Sea, through the Suez Canal, and around the Arabian Peninsula. This region is both data-scarce and source-rich, especially in the Persian Gulf where there are extensive Oil & Gas (O&G) activities. Analyses of the data presented in the manuscript lead to interesting findings about sources that contribute to observed hydrocarbon levels, as well as the importance of various chemical removal processes such as reactions with the hydroxyl and chlorine radicals. I recommend the paper be accepted after minor revisions are made to address the following comments.
* * *
**RC**: 1. The authors need to consider more carefully the precision of measured data and calculated ratios. It is not reasonable to report measured values with 4 or 5 digits of precision given what the authors have written about uncertainties in the measurement methods (see first paragraph on page 6). Based on what the authors have written there, I would say two digits of precision could be readily justified, but no way should values be reported with four or five significant figures.

Table 2 requires rethinking and extensive revisions to report all of the tabulated results with defensible precision. This issue also arises on page 9: 13.33 ppb of ethene at line 6 and similarly in numerous instances between lines 11 and 19 on the same page. On page 10: I would write 0.81, 0.96 and 0.88 for the ERs, and at line 16, I would write 27 and 54 ppb. See also page 15, line 4, and page 16, line 33 (108220:1, which appears again on line 5 of page 41 in the last figure caption.

**AC**: We agree with the reviewer that 4 or 5 digits of precision are not needed. However, some of the tabulated averages do justify the 3 digit of precision as the regional precisions were occasionally low enough to explain the reposted values.

**Changes made**: We have followed the suggested revisions of all values mentioned above. In addition, Table 2 is revised, reporting values that are justified by the regional precision. This approach has resulted in most of the values being reported with the 2nd digit of and some of them with the 3rd. For consistency, we have additionally changed the ppt reported values to ppb with the justified precision of two digits.
* * *
**RC**: 2. While some Arab governments prefer to use the term "Arabian Gulf", the main body of water between Iran and Saudi Arabia is still known internationally as the Persian Gulf. The authors should either use "Persian Gulf" or at least note the equivalence between Arabian Gulf and Persian Gulf in the text to reduce potential for confusion among readers. On page 7, line 20, Suez Gulf is typically referred to as the Gulf of Suez.

**AC**: We have incorporated the suggested terminology.

**Changes made**: We include in the abstract that the Arabian Gulf is "(*also known as Persian Gulf*)" and revised the Suez Gulf as "Gulf of Suez".
* * *
**RC**: 3. On page 9, line 5, the high ethene levels are linked to O&G activities, but to my knowledge O&G activities do not emit olefins. There can be a petrochemical source of ethene (e.g., in Houston) for example when ethane is converted to ethene and subsequently polymerized to make polyethylene. Also ethene is present in vehicle exhaust emissions.

**AC**: This is a good point raised by the referee, thank you. We now make it clear that the high ethene concentrations observed in the Gulf of Oman are not a result of direct emissions by O&G activities.

**Changes made**: This sentence now reads "*…indicated indirect influences from O&G production (i.e. dehydrogenation of ethane) in addition to urban pollution that includes vehicle exhausts and of marine traffic*".
* * *
**RC**: 4. Page 11, line 25, do you mean O&G instead of O&N here?

**AC**: We thank the referee for noticing this typo.

**Changes made**: Corrected to O&G.
* * *
**RC**: 5. Page 13, lines 13 and 33, do you mean excess mole fraction? The meaning of "the mole fraction of this sample" is unclear.

**AC**: We thank the referee for noticing the need to make this sentence clearer.

**Changes made**: We have specified that we are referring to the "*excess mole fraction of this sample*".
* * *
**Anonymous Referee #2**

**General evaluation**: This study reports the observed mixing ratios and relevant enhancement ratios for simple hydrocarbons including the light alkanes, alkenes, and C6-C8 aromatics via shipborne measurements conducted around the entire Arabian Peninsula on two separate transects. The authors collected high-quality data in an understudied region that contains many diverse emission sources including shipping, urban, oil and natural gas operations as well as very clean conditions with signatures of biogenic marine sources. Each of these sources was carefully identified using literature values and back trajectories to help determine likely source regions. The authors investigated how various emission source profiles and/or atmospheric oxidation would impact the observed enhancement ratios and variability in observed mixing ratios.

This was a well-written manuscript. I recommend publication after a few small changes/clarifications.

Technical comments in order of appearance:
* * *
**RC**: P1, L14 – Specify "This region…" as the Middle East, Arabian Gulf, etc., which ever term is most appropriate.

**Changes made**: We have specified "This region" with the most appropriate term of "*The Middle East*".
* * *
**RC**: P3, L30 – "…uses a dual-stage pre-concentration principle, additionally equipped with a focusing trap and a stripper column…" It was unclear to me if the focusing trap and the stripper column were part of the "pre-concentration principle" or occur after the sample pre-concentration. What exactly does the stripper column do? Does it remove the permanent gases, the lightest alkanes, etc.? A little more information would make this clear to the readers.

**AC**: We recognize the confusion that may arise for this sentence and the term "stripper column" since we are referring to the chromatographic column that is not a part of the pre-concentration system.

**Changes made**: We have revised the sentence as follows: "*Their main difference is that the GC5000VOC system uses a dual-stage pre-concentration principle, additionally equipped with a second trap (focusing trap / Carbosieve/Carbograph 1:1) and a second chromatographic column (AMA-sep WAX, 0.32 mm ID, 30 m, 0.25 µm) in order to…*".
* * *
**RC**: P4, L17 – I would state the differences as factors as it is easier to quickly comprehend (is 120% = a factor of 2.2 or 1.2?). Be sure to include the sign of the "difference," i.e.., positive or negative artifacts.

**AC**: This is a valid point as the increase or decrease is not clear for the reader.

**Changes made**: The sentence now reads "*Moist air (relative humidity 100 %) substantially influenced ethene concentrations compared to dry air (increased by a factor of 2.2) while it had a smaller effect on propene (increased by a factor of 1.16), benzene (increased by a factor of 1.28), and all other species (differences below $\pm$ 10 % between dry and moist air).*"
* * *
**RC**: P7, L22 – Please clarify what "this geographical demarcation" means. Do you mean "each region"? Perhaps, consider combining Figures 1 and 2 for ease of comparison with this statement.

**Changes made**: We have revised this sentence as follows: "*Despite the leg-to-leg differences in the origin of the air masses (Fig. 1), ethane mixing ratios (Fig. 2) match well with this geographical demarcation that has resulted in eight different regions.*"
* * *
**RC**: P8, L1 – Curious! Did you observe a diurnal profile in ethene in this region, increasing with daylight hours? The marine boundary layer height usually doesn't change much over open water, so a diurnal profile (particularly higher daytime concentrations) would bolster your hypothesis of a photo-sensitive marine biogenic source.

**AC**: We thank the referee for pointing out potential analysis that would have bolstered our hypothesis. Unfortunately, the data gaps during the first leg (Fig S13), and decreased precision due to extremely high waves in the second leg did not allow us to confirm our hypothesis through a clear diurnal cycle. What we have observed was mostly a hotspot rather than a uniform emission source around the Arabian Sea.

**Changes made**: We have removed the sentence that included the above hypothesis.
* * *
**RC**: P8, L7 – "Interestingly" appears five times and starts to become a bit redundant after the second time.  Please consider using sparingly, especially since it is a subjective term.

**Changes made**: We have removed "Interestingly".
* * *
**RC**: P8, L14 – Why would you "account for both isomers" when referring to the butanes? I'm not sure what is gained from that.

**AC**: We have accounted for both isomers to emphasize the high abundance of butanes (ca. 50%) over the Suez Canal. This is actually a unique feature of this region and we believe that this sentence serves the purpose to emphasize such abundance while it additionally re-directs the reader to the supplementary material.

**Changes made**: As the individual i- and n-butane average mixing ratios are reported in the previous line, we have kept the sentence in the manuscript.
* * *
**RC**: P8, L29 – Figure S20 does not show that the benzene to toluene ratio only the distribution of the mixing ratios. You could easily show this in a separate graph. Please include references or how you came to the conclusion that benzene/toluene > 1 indicates fresh ship emissions and/or biomass burning.

**AC**: We thank the referee for the suggestion.

**Changes made**: We have now included the review study of Andreae and Merlet (2001) and the most recent review of Andreae (2019) that shows benzene/toluene >1 is a characteristic signature for biomass burning plumes.
* * *
**RC**: L32-33 – Please list what the "marine traffic associated gases" are or refer to Figure6 to be discussed in section 3.2.1.

**AC**: We thank the referee for the suggestion.

**Changes made**: We now refer to Fig. 6 where our own ship exhaust composition of NMHCs is illustrated.
* * *
**RC**: P13, L12 – A pie chart similar to the ship emissions detailing the composition of this sample would be great to include, even in the SI.

**AC**: We thank the referee for the suggestion.

**Changes made**: The composition of this particular sample is now included in the supplementary information (Fig. S21).

[Figure]

**propane**
**n-butane**
**ethane**
**i-butane**
**n-pentane**
**i-pentane**
**n-hexane**
**2-methylpentane**
**m-,p- xylenes**
**n-heptane**
**benzene**
**toluene**
**octane**
**propene**
**2,2,4-Trimethylbenzene**
**trans-2-butene**
**1-pentene**

Fig. S21. Composition of the sample that was taken above the oil silk.
* * *
**RC**: P13, L13-14 – The authors state that it is a crude oil slick (L11), but are now calling it "associated gas." Also, beware of the "weathering effect" of oil slicks as the most volatile species tend to evaporate first and do not necessarily represent the actual composition of the starting material.

**AC**: We have defined as associated gas as the gas vapors that are emitted from liquid oil and therefore the oil slick can be considered as an example of associated gas. However, we appreciate the comment/feedback on the "weathering" effect and report on it.

**Changes made**: At the end of that paragraph we have added the following sentence: "*The composition of this sample (see Fig. S21) indicates a high fraction of light alkanes, but considering that the most volatile species tend to evaporate first, we cannot establish the representativeness of the actual composition of an oil slick*".
* * *
**RC**: P31 –Include the region names as you did with the previous table.  Also, are propane and methane really tracer compounds?  If so, of what?  Best to keep it simple and just state that you are presenting the VOC to propane or methane observed enhancement ratios.

**Changes made**: Table 3 now includes the complete names of regions. As suggested, the revised legend reads as "*Correlation matrix of enhancement volume mixing ratio slopes for selected alkanes and aromatics with propane and methane mixing ratios*".
* * *
**RC**: P38 – Ethane was lower inside the ship plume than the background values?

**AC**: That is indeed the case for the reported sample. But as stated in the text, the values fall inside the uncertainty limits. Nonetheless, this observation is representative of the frequently reported manuscript finding, that ethane is not emitted by marine traffic.
* * *
**RC**: P41 – Include what the picture is of and any photo credits.  I'm assuming it's of THE oil slick, not just a random oil slick, but I'm confused as it looks like barbed wire is in the foreground (?).  This makes me think that the picture was not taken from the ship. Perhaps best to simply leave out.

**AC**: The picture was indeed taken from the ship during the sampling of the particular sample. For satisfying the curiosity of the referee, the barbed wire served as protection from pirate attacks. Photo credits are not necessary as the picture was taken by the first author, E. Bourtsoukidis.

**Changes made**: We have included the following statement in the legend of the: "*The photo shows the oil slick as seen from the deck during the particular sample.*"
* * *
**RC**: P44 – Check the caption.  Replace pentane with butane.  There are also red bars for case study #1.

**AC**: We thank the reviewer for noticing this.

**Changes made**: We have now replaced the pentane with butane and also explain the red bars for case study #1.